DOI: 10.1038/s41467-017-01057-7　　**OPEN**

# Three-dimensional context rather than NLS amino acid sequence determines importin α subtype specificity for RCC1

Rajeshwer S. Sankhala[1], Ravi K. Lokareddy[1], Salma Begum[1], Ruth A. Pumroy[1,2], Richard E. Gillilan[3] & Gino Cingolani[1,4]

Active nuclear import of Ran exchange factor RCC1 is mediated by importin α3. This pathway is essential to generate a gradient of RanGTP on chromatin that directs nucleocytoplasmic transport, mitotic spindle assembly and nuclear envelope formation. Here we identify the mechanisms of importin α3 selectivity for RCC1. We find this isoform binds RCC1 with one order of magnitude higher affinity than the generic importin α1, although the two isoforms share an identical NLS-binding groove. Importin α3 uses its greater conformational flexibility to wedge the RCC1 β-propeller flanking the NLS against its lateral surface, preventing steric clashes with its Armadillo-core. Removing the β-propeller, or inserting a linker between NLS and β-propeller, disrupts specificity for importin α3, demonstrating the structural context rather than NLS sequence determines selectivity for isoform 3. We propose importin α3 evolved to recognize topologically complex NLSs that lie next to bulky domains or are masked by quaternary structures.

[1] Department of Biochemistry and Molecular Biology, Thomas Jefferson University, 233 South 10th Street, Philadelphia, PA 19107, USA. [2] Department of Biochemistry, University of Utah, 15N Medical Drive East, Salt Lake City, UT 84112-5650, USA. [3] Macromolecular Diffraction Facility, Cornell High Energy Synchrotron Source (MacCHESS), Cornell University, 161 Synchrotron Drive, Ithaca, NY 14853, USA. [4] Institute of Biomembranes and Bioenergetics, National Research Council, Via Amendola 165/A, Bari 70126, Italy. Correspondence and requests for materials should be addressed to G.C. (email: gino.cingolani@jefferson.edu)

Eukaryotic cells maintain an intracellular gradient of the small GTPase Ran that is predominantly bound to GTP in the nucleus and GDP in the cytoplasm[1–3]. This gradient is generated by the asymmetric distribution of Ran effectors: the guanine exchange factor RCC1 (regulator of chromatin condensation 1) is sequestered in the nucleus bound to chromatin, while the GTPase-activating protein specific for Ran, RanGAP, and various accessory Ran-binding proteins (RanBPs) reside in the cytoplasm. The Ran gradient promotes active, signal-mediated trafficking of macromolecules through the permeable barrier formed by nuclear pore complexes (NPCs) and is also essential for mitotic spindle assembly and nuclear envelope formation, all defining features of eukaryotic cells[4].

Cytoplasmic cargos bearing a nuclear localization signal (NLS cargos) are escorted through the NPC by soluble transport factors of the importin β-superfamily (or β-karyopherins), which share a conserved N-terminal Ran-binding domain[5]. β-karyopherins like the prototypical importin β[6] move through the NPC by interacting with phenylalanine-glycine (FG) repeats exposed by many nucleoporins in a RanGTP-dependent manner[7]. Importin β can recruit NLS cargos directly[8] or via the adaptor importin α[9], as in the 'classical' import pathway. The association between importin α and β is mediated by the N-terminal importin β-binding (IBB) domain[10] of importin α, which also binds NLS cargos via a C-terminal helical core composed of 10 Armadillo repeats (the 'Arm-core')[11]. Two NLS-binding pockets known as 'major' and 'minor' have been identified on the concave surface of importin α, spanning Arm repeats 2–4 and 6–8, respectively[11–14]. Classical monopartite NLSs like the SV40 T-antigen NLS generally bind at the major NLS binding pocket and, to a lesser extent, at the minor site, while bipartite NLSs like the nucleoplasmin NLS or membrane protein NLSs[15, 16] span both sites. In addition, several monopartite NLSs have been shown to preferentially[17] or exclusively[18–20] bind the minor NLS pocket of importin α. Interestingly, the human genome encodes seven isoforms of importin α, divided into three subtypes (or subfamilies), of which importin α1 functions as the generic adaptor for NLS cargos[21–24]. A growing number of specialized cellular NLS cargos like RCC1[24–27], STAT1[28], NF-kB[29, 30] as well as viral factors influenza PB2[31] and Ebola VP24[32] enter the nucleus in complex with specific importin α isoforms. These isoforms play an important role in cell differentiation[33], disease states, especially cancer[21] and viral infections[21, 34].

RCC1 is the founding member of the RCC1 superfamily that includes diverse cell cycle regulators containing one or more RCC1-like domains[35]. RCC1 consists of a C-terminal seven-bladed β-propeller[36] and N-terminal regulatory tail, which harbors a bipartite NLS[25, 26, 37] and a chromatin binding domain[38–40]. RCC1 can be imported into the nucleus in a classical Ran-dependent pathway that preferentially uses the isoform 3 of importin α[24, 25] and in a Ran-independent pathway that is not dependent on passive diffusion[26]. The RCC1 tail is post-translationally modified throughout the cell cycle. Methylation of the N-terminal Ser/Pro2 on its α-amino group by the α-N-methyltransferase NRMT[41] is important for high affinity binding to chromatin[39]. Mitotic phosphorylation of Ser11 by Cdc2 kinase reduces interaction with importin α3/β[42, 43], elevating the concentration of RanGTP on mitotic chromosomes and promoting binding of the RCC1:Ran complex to mitotic chromosomes, which is required for spindle assembly and chromosome segregation[44]. RCC1 interaction with chromatin is dynamic: RCC1 continuously binds to and dissociates from chromosomes to promote GTP exchange and this activity is dependent on its N-terminal tail[43]. Deletion of the RCC1 basic tail prevents RCC1 interaction with chromatin from being stabilized by Ran[45]. Likewise, Ran association with RCC1 regulates binding to both histones and DNA possibly by inducing an allosteric change in the tail that facilitates association of the binary Ran:RCC1 complex with chromatin[45, 46].

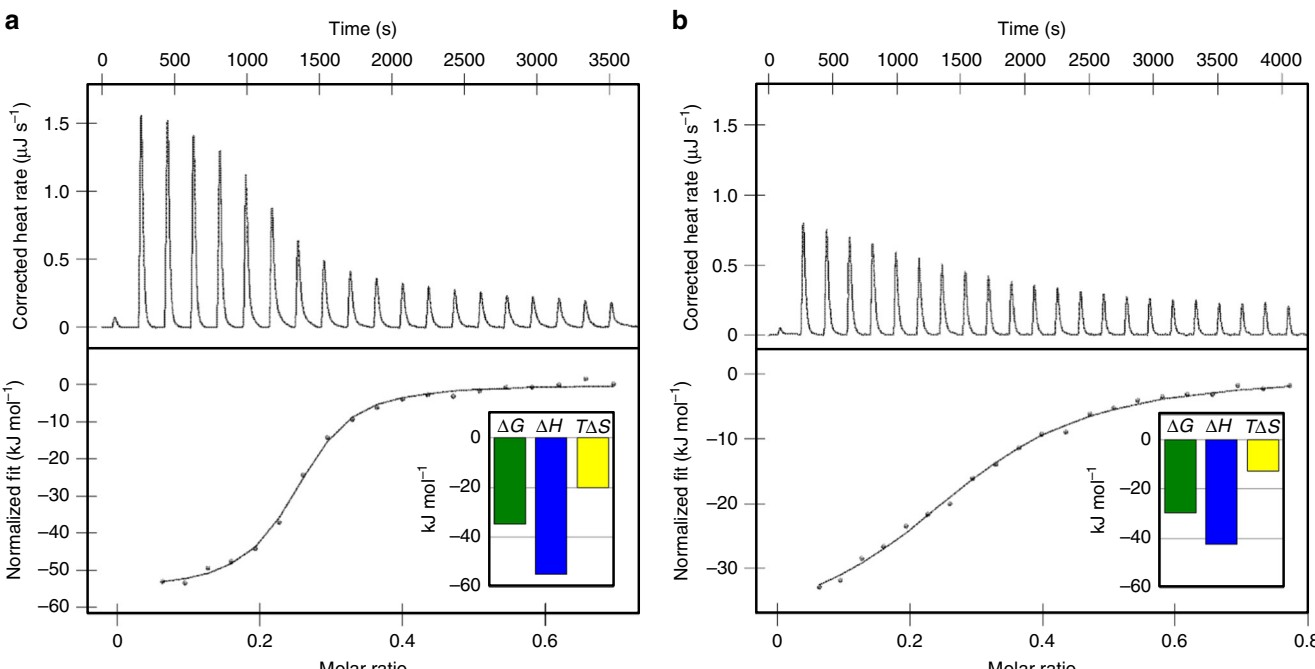

**Fig. 1** Calorimetric analysis of the interaction of human RCC1 with importin α isoforms. Titration of 300 μM RCC1 (in syringe) in a cell containing 100 μM **a** ΔIBB-importin α3 or **b** ΔIBB-importin α1. *Top panel*: raw injection heats. *Bottom panel*: integrated, buffer-subtracted binding enthalpy plotted as a function of the RCC1:importin α molar ratio. *Bottom panel insert*: overall variation of enthalpy (ΔH), entropy (TΔS), and Gibbs (ΔG) energy associated with each binding titration

**Table 1 Crystallographic data collection and refinement statistics**

|  | ΔIBB-importin α3: RCC1 | IBB-Kap60: yRCC1 |
|---|---|---|
| *Data collection* |  |  |
| Space group | P1 | P2$_1$2$_1$2$_1$ |
| Cell dimensions |  |  |
| $a$, $b$, $c$ (Å) | 127.4, 162.5, 161.6 | 91.1, 99.2, 125.0 |
| $\alpha$, $\beta$, $\gamma$ (°) | 75.7, 85.6, 72.2 | 90.0, 90.0, 90.0 |
| Wavelength (Å) | 0.97 | 1.18 |
| Resolution (Å) | 50–3.45 (3.53–3.45) | 50–2.63 (2.72–2.63) |
| No. of reflections (tot/unique) | 935,054/143,482 | 160,524/33,934 |
| $R_{sym}$ | 13.1 (65.5) | 4.9 (64.4) |
| $R_{pim}$ | 11.2 (65.9) | 2.2 (42.8) |
| $I/\sigma I$ | 12.2 (1.8) | 49.9 (3.4) |
| Completeness (%) | 90.4 (71.5) | 96.0 (85.8) |
| Redundancy | 2.0 (1.6) | 4.7 (2.7) |
| *Refinement* |  |  |
| PDB ID | 5TBK | 5T94 |
| Resolution (Å) | 20–3.45 | 15–2.63 |
| No. of reflections | 118,199 | 31,489 |
| $R_{work}/R_{free}$* | 27.8/29.6 | 21.7/25.9 |
| No. of complexes in AU | 8 | 1 |
| No. of protein atoms | 50,677 | 6808 |
| Ramachandran |  |  |
| Favored/allowed/outliers | 96.1/3.9/0.0 | 95.3/4.6/0.1 |
| MolProbity Clashscore | 8.8 | 11.6 |
| R.m.s. deviations |  |  |
| Bond lengths (Å) | 0.005 | 0.002 |
| Bond angles (°) | 0.684 | 0.496 |

Values in parentheses are for highest-resolution shells
*$R_{free}$ was calculated using ~5% randomly selected reflections

In this paper, we have determined the mechanisms by which RCC1 is selectively recognized and translocated into the cell nucleus by importin α3.

## Results

**Structure of the importin α3:RCC1 complex.** RCC1 is preferentially imported by importin α3, though in vitro it can also bind the universal importin α1[25, 26]. Using nano isothermal titration calorimetry (ITC), we established that isoform 3 has one order of magnitude higher binding affinity for RCC1 than importin α1 ($K_D = 0.62 \pm 0.1\,\mu M$ vs. $5.4 \pm 0.3\,\mu M$) (Fig. 1a, b). To shed light on the structural basis for importin α3 selectivity, we crystallized the predominant isoform of human RCC1 (or RCC1α[47]) in complex with importin α3 lacking the IBB-domain. Large crystals containing eight copies of the α3:RCC1 complex in the asymmetric unit were obtained in the triclinic space group. The structure was solved by maximum likelihood molecular replacement using individual atomic models of ΔN-RCC1[36] and ΔIBB-importin α3[31]. Flexible torsion-angle non-crystallographic symmetry restrains[48] among the eight copies in the asymmetric unit were used throughout refinement. This yielded an accurate atomic model ($R_{work}/R_{free}$ ~27.8/29.6) (Table 1), despite the modest resolution of crystallographic data (~3.45 Å). The importin α3:RCC1 complex (Fig. 2a) is shaped like a number 'nine'. Clear electron density was observed for the RCC1 N-terminal tail, which is missing in our phasing model, and contains a bipartite NLS between residues 4 and 26 (Fig. 2b). The tail occupies the entire NLS-binding groove of importin α3, spanning ~50 Å between Arms 2 and 9 (Fig. 2a). It continues into a seven-bladed β-propeller (residues 30–421) that packs against

the lateral surface of importin α3, making minimal contacts with Arms 1–4. The β-propeller has a disk-like structure with two faces: one face that binds Ran[49] is solvent exposed, while the opposite face contacts the C-terminal portion of the tail (residues 21–29), including the major NLS box, which is sandwiched between the importin α3 helical groove and the β-propeller (Fig. 2a). Significant structural differences (RMSD 2.1 Å) exist in the eight importin α3:RCC1 complexes in the triclinic cell (Supplementary Fig. 1a), although there is no obvious correlation between crystal contacts and the RMSD of RCC1 protomers crystallized in the P1 cell. Normal mode analysis[50] identifies a hinge-axis between the β-propeller and the N-terminal tail running along residues 23–27, which results in a hinge-bending movement of the β-propeller toward importin α3 of up to 8 Å.

To determine whether the eight importin α3:RCC1 complexes trapped in the crystal structure reflect a physiologically significant stoichiometry, we investigated the oligomeric state of the purified complex in solution using Small angle X-ray analysis (SAXS) (Fig. 2c). At physiological salt and in a range of concentrations between 3.5 and 7.5 mg ml$^{-1}$ (Supplementary Fig. 2a), the importin α3:RCC1 complex has a radius of gyration (RG) of $34.6 \pm 0.9$ Å (maximum diameter, $D_{max}$ ~115.3 Å) and a Porod volume of ~177,403 Å$^3$, consistent with a heterodimer in 1:1 stoichiometry (expected MW ~91.2 kDa) (Supplementary Table 1). The Kratky plot of SAXS data has a bell-shaped profile with a single pronounced maximum suggestive of a well-folded biopolymer[51] (Supplementary Fig. 2a). An ab initio shape reconstruction from merged scattering data revealed an asymmetric ellipsoidal shape, similar to a number 'nine' (Fig. 2d). We interpreted this envelope by docking an ensemble of the eight slightly different importin α3:RCC1 complexes observed in the triclinic unit cell, yielding a very good agreement between observed and calculated SAXS data ($\chi^2 = 1.6$) (Fig. 2c). The hinge region between the β-propeller and N-terminal tail is larger in the SAXS envelope than in the crystallographic model, reflecting a flexible connection between these domains (Supplementary Fig. 1a). The SAXS envelope is also slightly more elongated than the crystallographic model, consistent with the notion that importin α3 stretches in solution[31].

**Importin α3:RCC1 binding interface.** Two of the eight complexes in the asymmetric unit have better electron density and were used as references for structural analysis. In these two complexes, RCC1 buries 1880.8 Å$^2$ of the exposed surface of importin α3: 23 residues of RCC1 interact with 39 residues of importin α3, which account for a predicted binding energy of $-12.6$ kcal mol$^{-1}$. RCC1 makes contacts with three regions of importin α3, schematically illustrated in Fig. 3a. The first region includes the RCC1 minor NLS box, which is remarkably extended in this import cargo and includes three basic residues (8-KRR-10) bound at positions P$_1'$–P$_3'$, plus two additional basic residues (4-KR-5) at P$_{-3}'$ and P$_{-2}'$ (Supplementary Table 2). The critical S11, which is phosphorylated during mitosis to reduce association of RCC1 with importin α3[42, 43], occupies position P4'. The second region of contacts includes RCC1 residues (20-SKKVK-24) that occupy the major NLS-binding pocket at positions P$_1$–P$_{45}$, with K21 at the crucial P$_2$ site. This NLS is somewhat different than classical NLSs[52]: it contains a basic residue (K19) at position P$_0$ and two non-basic residues at P$_1$ and P$_4$ (S20 and V23) with K24 at P$_5$ making extensive contacts with importin α3 (Fig. 3a). The portion of the RCC1 tail where the major NLS lies is sandwiched between importin α3 helical core and the β-propeller that possibly destabilizes the interaction with α3. Accordingly, the refined B-factor of RCC1 atoms is lower at the minor NLS box than at the

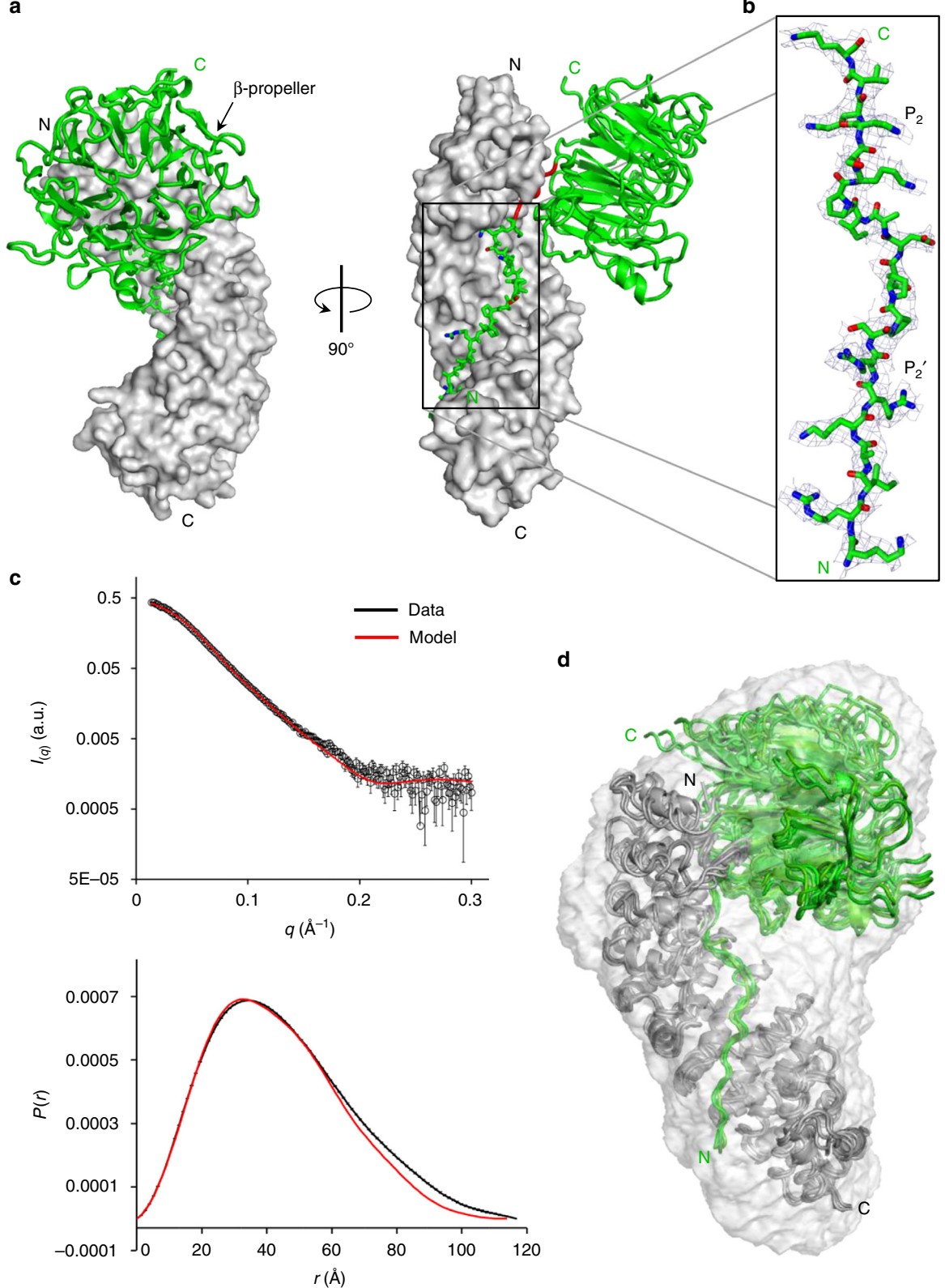

**Fig. 2** Structure of human importin α3 bound to RCC1. **a** A representative structure of the human importin α3:RCC1 complex with RCC1 (ribbon diagram) and importin α3 (solvent surface) colored in green and gray, respectively. RCC1 hinge residues (23–27) are colored in red. **b** Eight-fold non-crystallographic symmetry averaged 2Fo − Fc electron density map of the RCC1 NLS displayed at 1σ above background. The density (in *blue*) is overlaid to residues 4–24 of the final refined model (in *green*). **c** SAXS analysis of the importin α3:RCC1 complex. Experimental scattering data (shown in *black*) obtained by merging scattering data at 3.5, 5.0, and 7.5 mg ml$^{-1}$ (*top panel*) and corresponding distance distribution function P(r) (*bottom panel*). Scattering profile and P(r) function calculated from the average of the eight importin α3:RCC1 complexes observed crystallographically are shown in red. **d** Ab initio SAXS reconstruction of the importin α3:RCC1 complex calculated from merged scattering intensities at 3.5, 5.0, and 7.5 mg ml$^{-1}$. Overlaid to the SAXS envelope is a composite of all complexes in the triclinic unit cell

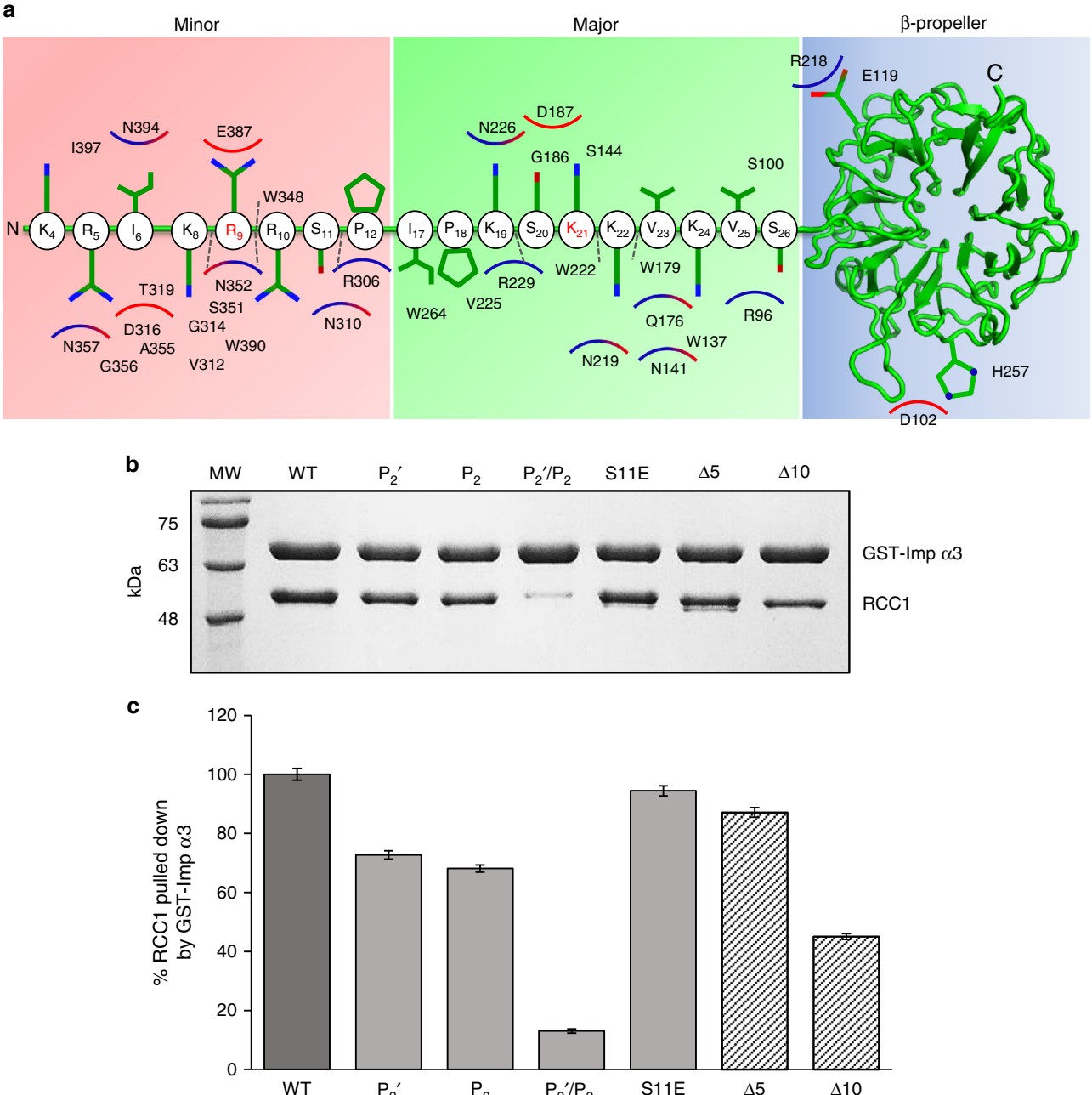

**Fig. 3** Importin α3:RCC1 binding interface. **a** Schematic diagram of interactions between RCC1 and importin α3. R9 and K21 at positions P$_2$′ and P$_2$, respectively, are shown as *red letters*. **b** Pull-down analysis of the interaction of GST-ΔIBB-importin α3 immobilized on glutathione beads with RCC1. Loading controls are in Supplementary Fig. 3a **c** Quantification of pull-downs shown as mean ± SD for three independent experiments. No interaction is observed between free RCC1 and glutathione beads (Supplementary Fig. 3d)

major site (85 vs. 105 Å$^2$), suggesting the minor NLS-binding site is the main site of interaction with importin α3. Finally, two residues in the β-propeller, E119 and H257 contact R218 and D102, respectively, in importin α3 (Fig. 3a) although the long bonding distance (3.5–5 Å) suggests a minor energetic contribution to the binding interface.

We used a quantitative pull-down assay to determine how mutations in the RCC1 NLS disrupt binding to importin α3 in vitro. Purified GST-tagged importin α3 (lacking the IBB) and RCC1 were incubated at physiological concentrations (~1 and 0.75 μM, respectively[53]) and after reaching equilibrium, the complex was immobilized on glutathione beads, washed to remove unbound RCC1 and eluted with reduced glutathione.

Identical elution volumes were analyzed by SDS-PAGE and quantified in triplicates. As expected, ΔIBB-importin α3 bound RCC1 in a 1:1 molar ratio (or 100% of importin α bound to cargo) (Fig. 3b, c). Six RCC1 mutants were generated and tested for binding to importin α3: three mutants that disrupt the P$_2$ and/ or P$_2$′ sites (RCC1-K21A; RCC1-R9A; RCC1-R9A/K21A); two deletion mutants that lack either the post-translationally modified N-terminal moiety or the entire minor NLS box (Δ5-RCC1 and Δ10-RCC1), and a phospho-mimetic RCC1-S11E, which mimics a phosphorylation at S11[42, 43]. We found that point mutations at either P$_2$ or P$_2$′ reduced RCC1-binding to importin α3 by 25% and combining these two mutations had a synergistic effect[54] reducing binding by ~85% (Fig. 3b, c). A phospho-mimetic

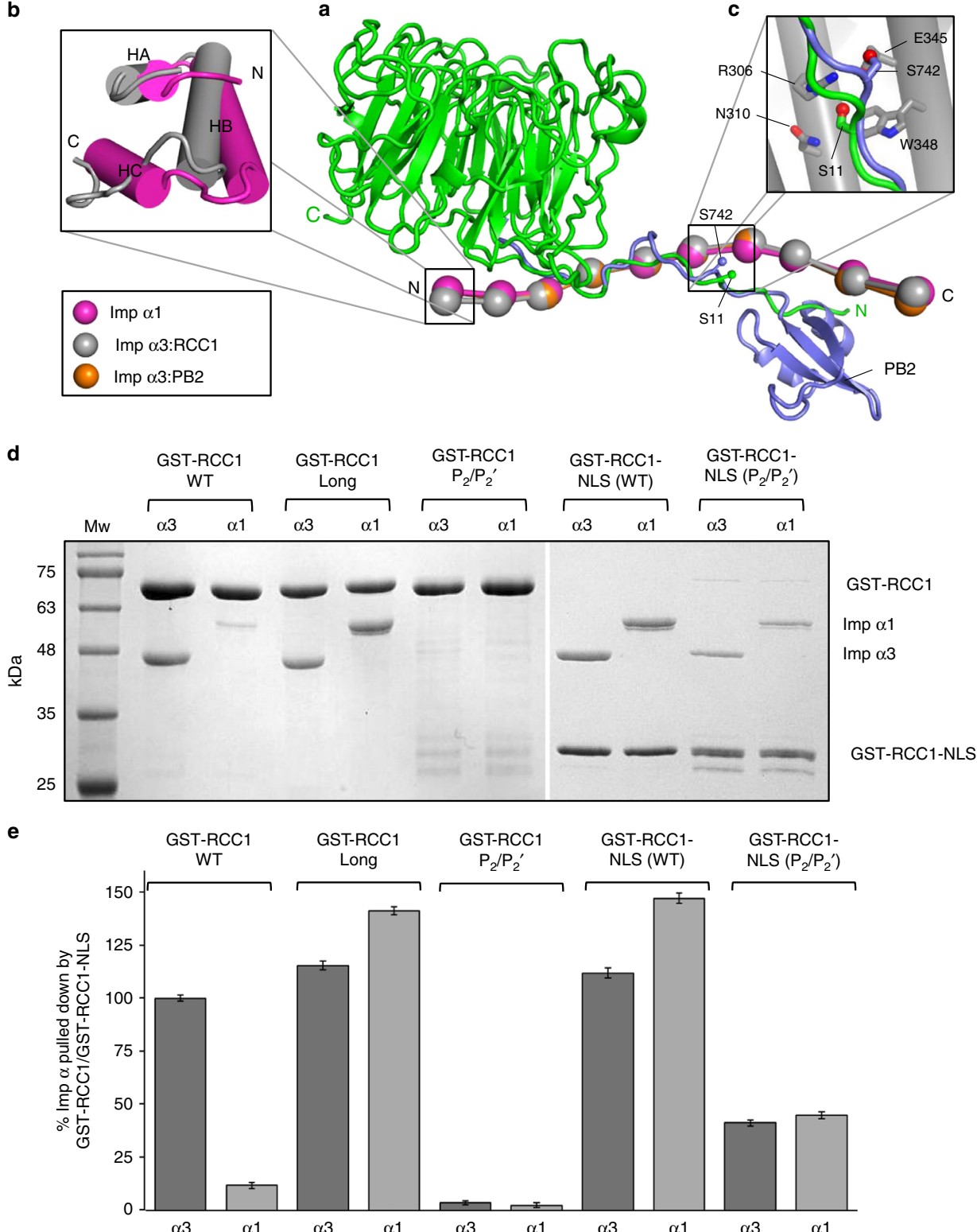

**Fig. 4** Structural flexibility in importin α3 promotes RCC1 recognition. **a** Structural superimposition of importin α3:RCC1 (pdb: 5TBK), importin α3:PB2 (pdb: 4UAE), and importin α1:nucleoplasmin NLS (pdb: 1EJY). Importin αs are shown as beads-on-a-string (each bead representing an Arm repeat[31]) while PB2 and RCC1 are displayed as ribbons; the nucleoplasmin NLS is omitted for clarity. **b** Zoom-in panel showing a superimposition of Arm 1 in importin α3 bound to RCC1 (*gray*) and importin α1 (*magenta*). **c** Zoom-in panel showing the local environment of the RCC1 S11 and PB2 S742, which are both subjected to phosphorylation. **d** Pull-down analysis of the interaction of importin α3 and α1 with the full-length GST-RCC1, GST-RCC1-Long, GST-RCC1-$P_2$/$P_2'$ or just the RCC1-NLS (e.g., GST-RCC1-NLS (WT) and GST-RCC1-NLS ($P_2$/$P_2'$)). The gel shows identical fractions eluted from glutathione beads. Loading controls are in Supplementary Fig. 3b. **e** Quantification of pull-downs in **d** shown as mean ± SD for three independent experiments. No interaction was observed between free importin α1/3 and glutathione beads (Supplementary Fig. 3e)

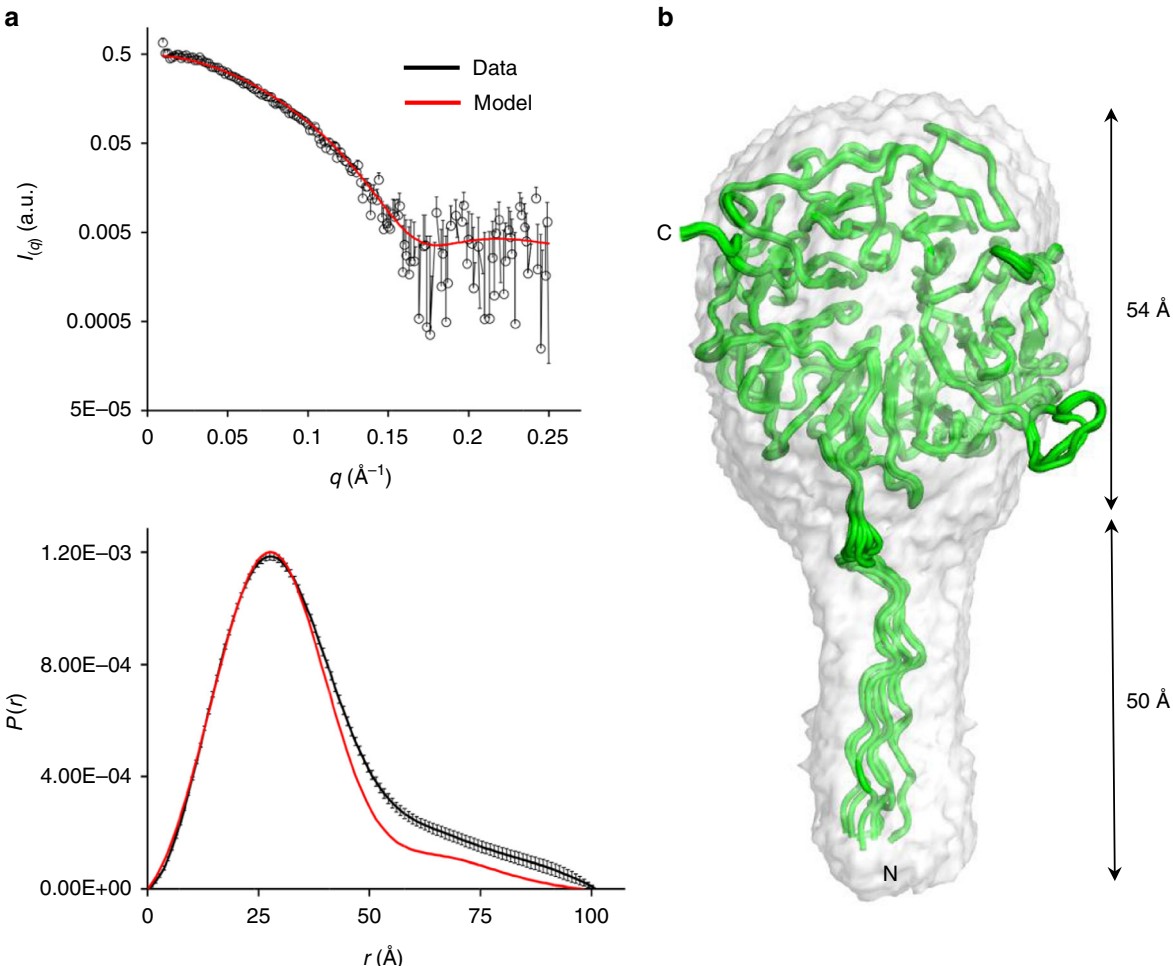

**Fig. 5** RCC1 N-terminal tail exists in a conformational ensemble. SAXS analysis of human RCC1. **a** Experimental scattering data at 2.5 mg ml$^{-1}$ (*top panel*) and *P*(*r*) function for observed SAXS data (*bottom panel*) are shown in black and data calculated from an *ensemble* model of eight RCC1s observed crystallographically and docked inside the SAXS envelope are shown in red. **b** Ab initio SAXS reconstruction of human RCC1 calculated from experimental scattering data at 2.5 mg ml$^{-1}$. Overlaid to the SAXS envelope are the eight conformers of RCC1 observed crystallographically in complex with importin α3

mutation at S11 or deletion of the first five residues (Δ5-RCC1) reduced association with importin α3 by ~10%. More pronounced (~60%) was the loss of importin α3 binding to Δ10-RCC1, which lacks R9 at position P$_2$′ as well as other important residues at the minor NLS box (Fig. 3b, c). Finally, deletion of the entire NLS completely disrupted RCC1 binding to importin α3 and Ran-dependent nuclear import of GFP-RCC1[26].

**Importin α3 conformational flexibility promotes RCC1 binding.** A structural superimposition of the universal importin α1 with importin α3 bound to RCC1 and PB2[31] reveals both α3-specific cargos have NLSs that run antiparallel to the Arm-core occupying similar positions in the NLS-binding groove (Fig. 4a). However, the RCC1 β-propeller projects toward the N terminus of importin α3 rather than the C-terminus, as the PB2 globular domain does. Importin α1 is essentially identical to importin α3 bound to PB2 (RMSD 1.01 Å) but differs from importin α3 in complex with RCC1 (RMSD 1.36 Å), which undergoes two local conformational changes at its N terminus (Fig. 4b): first, Arms 1–2 rotate ~20° away from the RCC1 β-propeller adopting a more open conformation than in α1; second, the helix C of Arm 1 unfolds in the importin α3:RCC1 complex, while it rests folded as a canonical α-helix in importin α1 and in the importin α3:PB2 complex. Modeling RCC1 bound to

importin α1 reveals severe clashes between Arms 1–2 and the β-propeller (Supplementary Fig. 1b), explaining the low micromolar binding affinity measured by ITC (Fig. 1b). On the contrary, the extended importin α3 solenoid, which has a hinge-bending region near Arms 3–4[31], allows the first three repeats to open up and accommodate the β-propeller while making close contacts with the bipartite NLS.

To validate this idea experimentally, we probed the association of importin α3 and α1 (lacking the autoinhibitory IBB) with either the full length GST-RCC1 or just the bipartite RCC1 NLS (GST-RCC1-NLS). A pull-down assay carried out using physiological concentrations of importin αs (~1 μM) and RCC1 (~0.75 μM)[53] confirmed importin α3 binds the full-length RCC1 significantly better than the isoform α1 (Fig. 4d, e). A stoichiometric quantity of isoform 3 was pulled-down by RCC1 vs. less than ~12% of importin α1, consistent with a ~10-fold lower $K_D$ measured by ITC (Fig. 1). In contrast, a slightly longer construct of RCC1 (RCC1-long) that bears a 12 residue linker (GSAGSAAGSGEF) between residues S26 and H27, right at the junction between the β-propeller and NLS, efficiently bound both importin α1 and α3, showing no selectivity for the isoform 3 (Fig. 4d, e). As expected, the interaction of RCC1 with either isoform was completely disrupted by a dual point mutation at P$_2$/P$_2$′. Unlike the full-length RCC1, the isolated RCC1 NLS bound either importin α isoform with high affinity and this

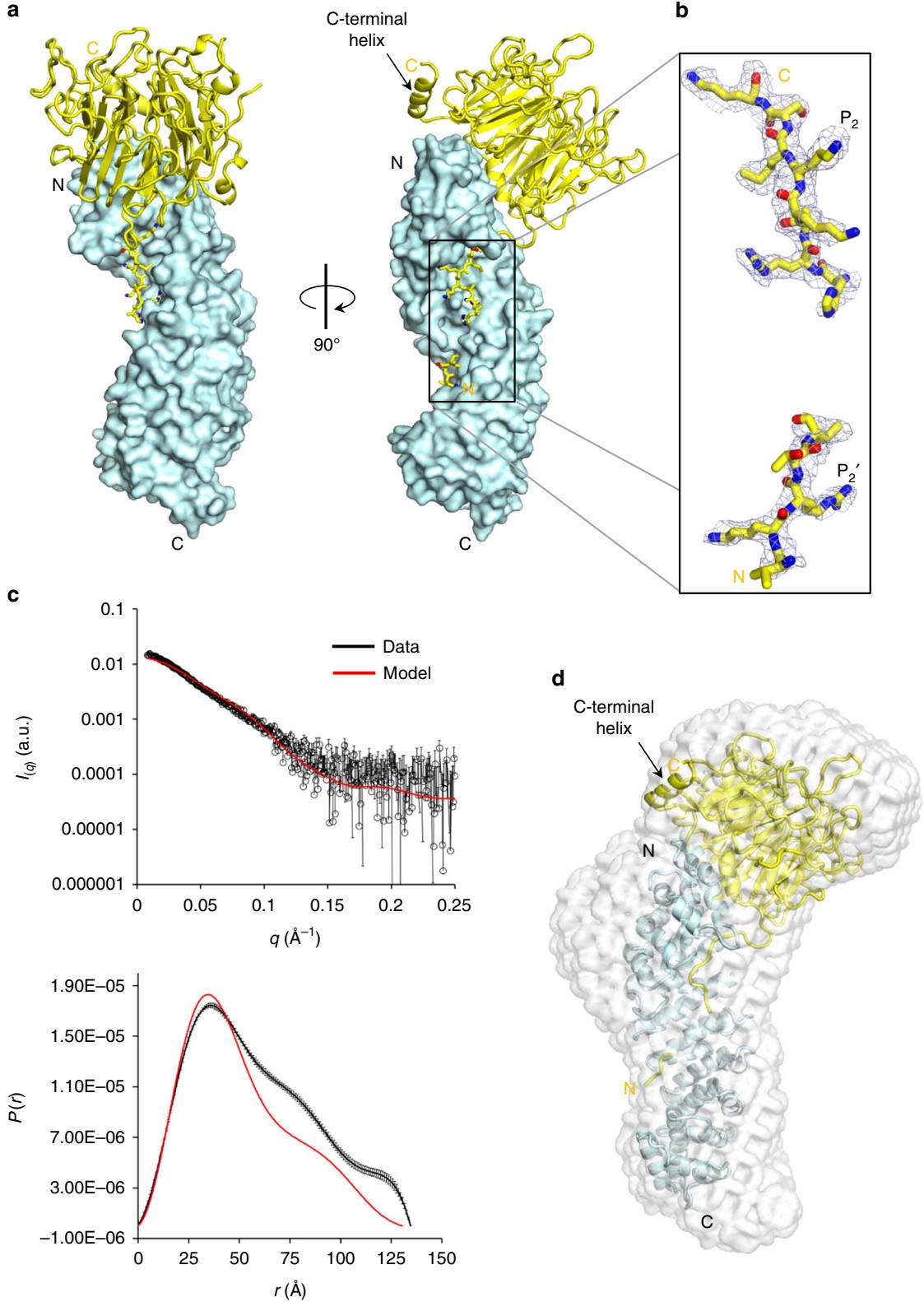

**Fig. 6** The structure of yeast RCC1 in complex with Kap60. **a** Crystal structure of Kap60 (*light cyan*) bound to yRCC1 (*yellow*). **b** A representative *F*o − *F*c electron density difference map of yRCC1 N-terminal tail (in *blue*) is displayed at 2σ above background and is overlaid to the final refined model (residues 2–6 and 16–23) colored in *yellow*. **c** Average SEC-SAXS data calculated from frames 710 to 750 (*top panel*) and corresponding *P*(*r*) function (*bottom panel*) are shown in black. Scattering data and *P*(*r*) function calculated from the crystal structure of the Kap60:yRCC1 complex are shown in *red*. **d** Ab initio SAXS reconstructions of the Kap60:yRCC1 complex calculated from experimental scattering values obtained from SEC-SAXS (frames 710–750). The crystallographic structure of the complex is overlaid to the SAXS envelope

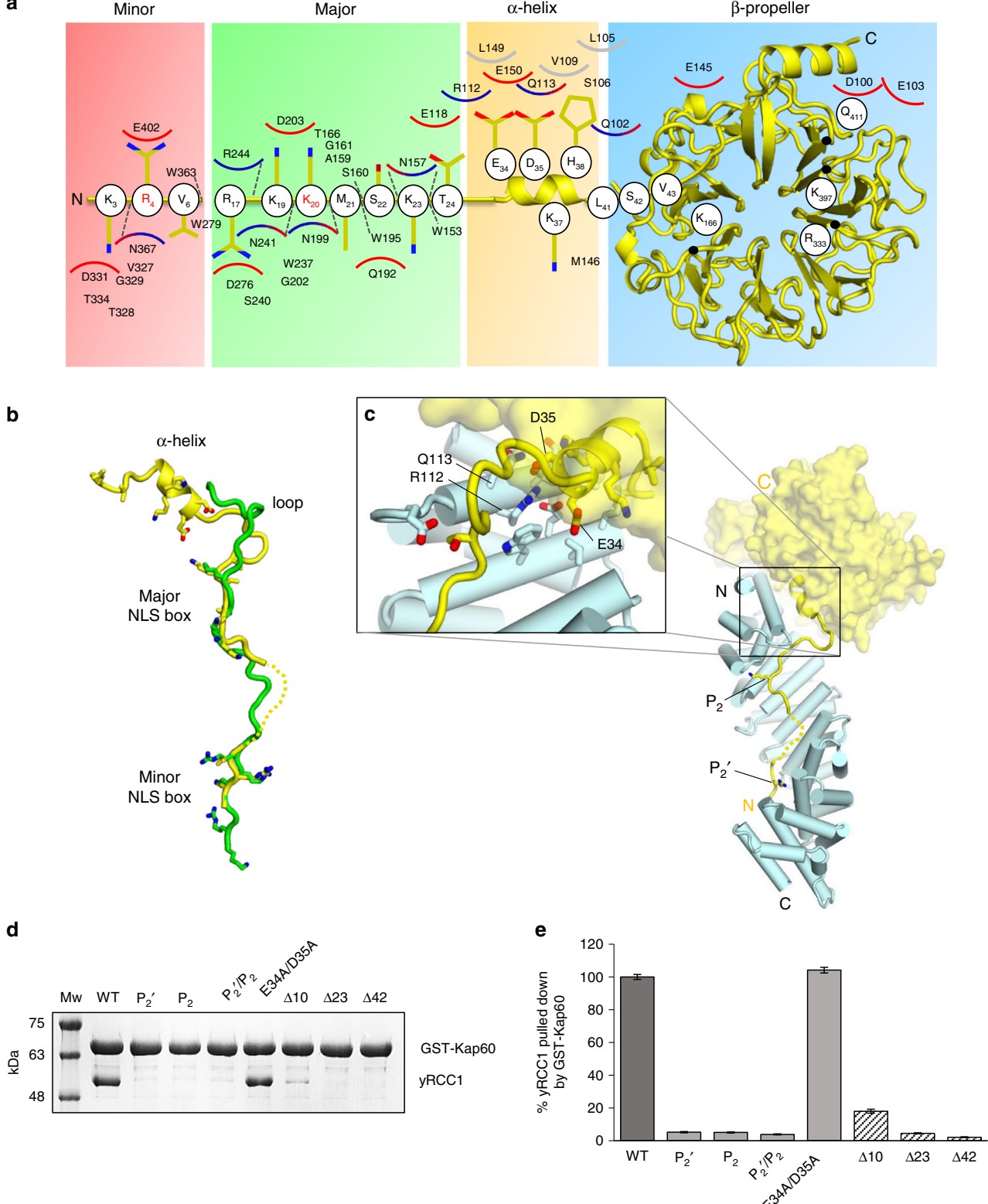

**Fig. 7** Kap60:yRCC1 binding interface. **a** Schematic diagram of the interactions between the yRCC1 N-terminal tail and Kap60. R4 and K20 at positions P₂' and P₂, respectively, are shown as *red* letters. **b** Superimposition of the bipartite NLSs and C-terminal regions of yeast and human RCC1 colored in *yellow* and *green*, respectively. **c** Crystal structure of Kap60 (*light cyan*) bound to yRCC1 (*yellow*) illustrating contacts between the C-terminal α-helix and Kap60 Arm 1. **d** Pull-down analysis of the interaction of GST-ΔIBB-Kap60 immobilized on glutathione beads with yRCC1. Loading controls are in Supplementary Fig. 3c. **e** Pull-downs are shown as mean ± SD for three independent experiments. No interaction was observed between free yRCC1 and glutathione beads (Supplementary Fig. 3d)

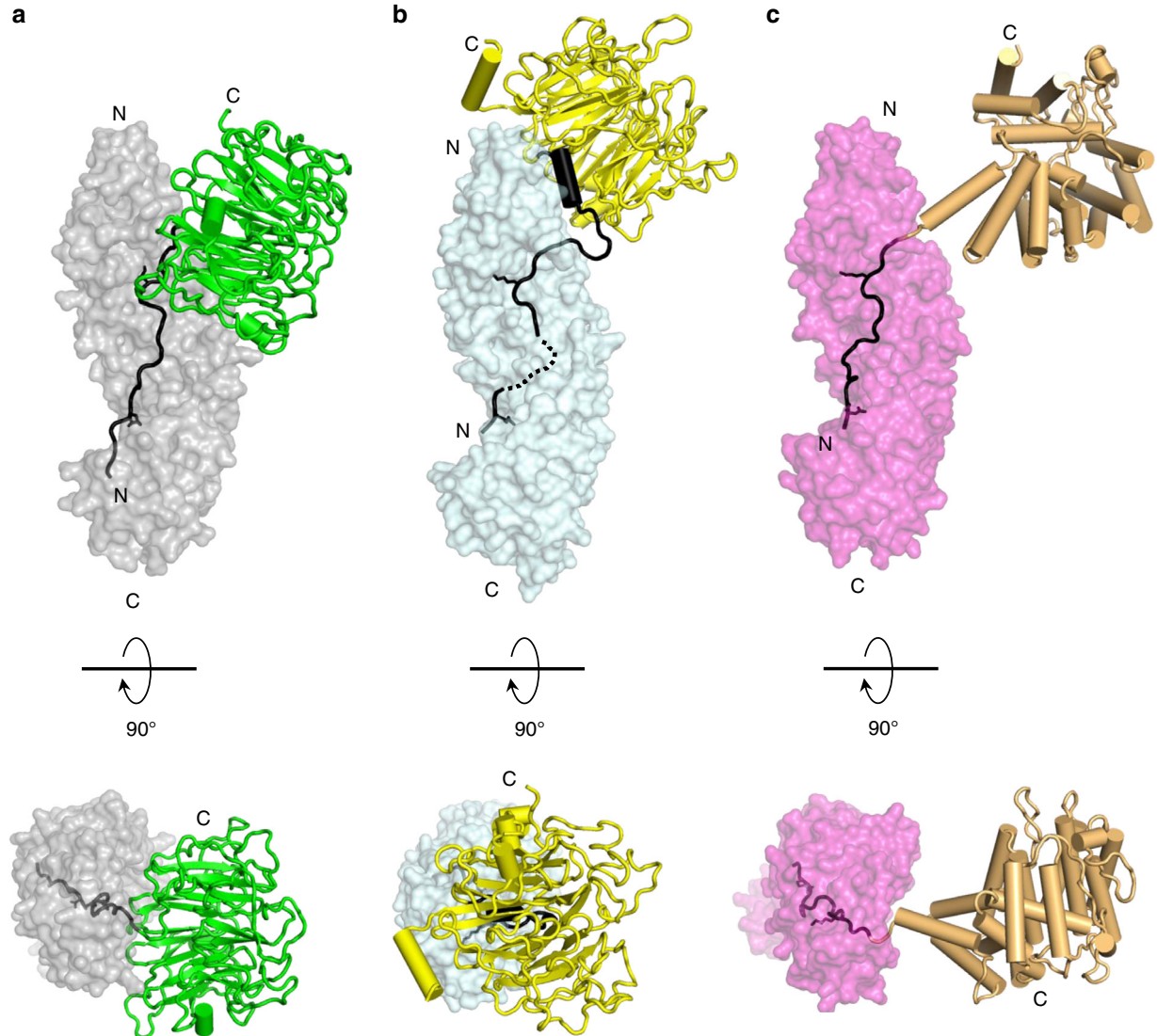

**Fig. 8** Differential recognition of NLS cargos by importin αs. Recognition of **a** human RCC1 (*green*) by importin α3 (*gray*), **b** yeast RCC1 (*yellow*) by Kap60 (*light cyan*), **c** CAP80 (*orange*) by importin α1 (*magenta*) (PDB ID 3FEY). For clarity, residues 385–790 of the CAP80 are not shown. NLSs and flanking residues that make contacts with importin αs are shown in black; residues at $P_2$ and $P_2'$ are shown as sticks

interaction was reduced, but not completely disrupted, by mutations at $P_2/P_2'$ (Fig. 4d, e). Thus, the enhanced affinity of importin α3 for RCC1 results from the ability of this isoform to bind the RCC1 NLS in the context of the full-length import cargo, which contains a bulky β-propeller domain flanking the NLS.

**RCC1 NLS samples multiple conformations in solution**. Intrigued by the extended conformation observed crystallographically, we investigated the structure of the full-length RCC1 in solution, in the absence of importin α3. SAXS analysis in a concentration range between 2.5 and 5.0 mg ml⁻¹ (Fig. 5a, Supplementary Fig. 2b) revealed a RG = 27.8 ± 0.7 Å and $D_{max}$ ~102.5 Å, similar to the total length of RCC1 co-crystallized with importin α3. Likewise, the Porod volume is ~100,732 Å³, which is also consistent with a monomer (expected MW ~49.0 kDa) (Supplementary Table 1). The Kratky plot has a slightly broad bell-shaped profile, suggestive of a flexible protein[51] (Supplementary Fig. 2b). The $P(r)$ function calculated from SAXS has a distinct symmetric peak indicative of a globular domain ~50 Å in diameter flanked by a less ordered moiety (Fig. 5a). An ab initio

shape reconstruction from scattering data (at 2.5 mg ml⁻¹) suggests RCC1 looks like a "tennis racket" (Fig. 5b). The β-propeller, ~44 Å in diameter, fits remarkably well inside the globular portion of this envelope with the extended (~50 Å) N-terminal tail occupying the "racket handle". The elongated shape suggested by SAXS was also validated by hydrodynamic analysis using analytical ultracentrifugation sedimentation velocity (AUC-SV) (Supplementary Table 1) that, at comparable concentration, yields a frictional ratio $f/f_0$ ~1.7, indicative of an elongated macromolecule. RCC1 N-terminal moiety is susceptible to proteolysis in solution and likely samples multiple conformations in the absence of importin α3. A molecular dynamics (MD) simulation using the crystallographic structure of RCC1 as the initial state revealed the N-terminal tail is the most dynamic region of RCC1 with root mean square fluctuation (RMSF) up to 5 Å² (Supplementary Fig. 4). Good agreement between observed SAXS data and the crystallographic model ($\chi^2 = 2.6$) was obtained using a docked ensemble of the eight RCC1 structures observed in the triclinic unit cell (Fig. 5b), aligned globally using the β-propeller as a template. In this ensemble model, the eight β-propellers occupy nearly identical positions (RMSD 0.9 Å), while the RCC1 tails are

displaced up to 10 Å with RMSD >4 Å between residues 1 and 7. Thus, the NLS of human RCC1 samples different conformations in solution in the absence of importin α3.

**The structure of yeast RCC1 in complex with Kap60.** Unlike humans that express seven isoforms of importin α, *Saccharomyces cerevisiae* has only one importin α gene encoding Kap60. This karyopherin is phylogenetically more similar to human importin α3/4 than α1 and falls in the α2-subfamily[21, 22] (Supplementary Fig. 5). To determine how Kap60 binds the yeast homolog of mammalian RCC1 (Prp20p, herein referred to as yRCC1), we co-expressed the two proteins in bacteria, purified a homogeneous 1:1 complex and obtained large orthorhombic crystals of the Kap60:yRCC1 complex, which diffracted X-rays to high resolution. The structure of the Kap60:yRCC1 complex was solved by molecular replacement and refined to an $R_{work}/R_{free}$ of 21.7/25.9%, at 2.63 Å resolution (Table 1, Fig. 6a). Difference maps revealed strong electron density at the N terminus of the β-propeller, which allowed us to model yRCC1 residues at the major and minor NLS-binding sites of Kap60 (residues 41–47 and 3–6, respectively) (Fig. 6b). Unexpectedly, in the yeast complex the β-propeller is positioned onto the surface of Arms 1–2 of Kap60, making the complex more elongated and slender than the human counterpart (~135 vs. ~115 Å) (Fig. 2a, Supplementary Fig. 6a). To validate the crystallographic structure, we attempted conventional SAXS analysis. However, the strong tendency of the Kap60:yRCC1 complex to aggregate, even at low concentration, precluded meaningful SAXS analysis. We then turned to size exclusion chromatography coupled SAXS (SEC-SAXS)[55] that allows to separate aggregates from monodisperse species during SAXS analysis. This analysis revealed an RG = 45.9 ± 0.5 Å and a $D_{max}$ ~137.0 Å (Fig. 6c, Supplementary Table 1), in good agreement with the crystallographic structure (maximum length ~135.0 Å). The Porod mass of the yRCC1:Kap60 complex calculated from SAXS data is ~99,000 kDa, also consistent with a 1:1 stoichiometry (expected MW ~103.0 kDa) (Fig. 6c, Supplementary Table 1), which was independently validated using AUC sedimentation velocity analysis (Supplementary Fig. 7g, h). Small differences between the $P(r)$ function calculated from SAXS data and computed from the crystallographic model persist in the data (Fig. 6c) and likely reflect an intrinsic tendency of the Kap60:yRCC1 complex to self-associate in solution forming aggregates. An ab initio shape reconstruction from SEC-SAXS data reveals an elongated shape (Fig. 6d), in reasonable agreement with the crystallographic structure ($\chi^2$ ~2.15). The SAXS envelope is however broader than the crystallographic model, especially in the region where yRCC1 was docked, possibly due to motion of the C-terminal helix that could swing by 180° toward the Arm core (Fig. 6d). Thus, SEC-SAXS analysis supports the crystallographic description of the Kap60:yRCC1 complex (Fig. 6a), whereby the yRCC1 β-propeller is positioned onto Arms 1–2 as opposed to leaning against the lateral surface of Arms 1–4, as seen in the importin α3:RCC1 complex (Fig. 2a).

**Kap60:yRCC1 binding interface.** Yeast RCC1 buries a large exposed surface of Kap60, comparable to the human RCC1: importin α3 complex (1916.2 vs. 1880.8 Å²). This binding interface is stabilized by 27 residues in yRCC1 that interact with 47 residues of Kap60, which in total account for a predicted binding energy of −12.8 kcal mol⁻¹ (Fig. 7a). The binding interface is more complex than seen for human RCC1 and can be divided into four regions. First, at the minor NLS-binding pocket, yRCC1 residues K3 and R4 make close contacts at positions $P_1'$ and $P_2'$ of Kap60 (Fig. 6b, Supplementary Table 2). Unlike the human complex (Fig. 2b), residues connecting major and minor NLS

boxes are invisible in the crystal structure, possibly due to the lack of specific contacts with Kap60. Second, at the major NLS pocket, seven yRCC1 residues make extensive contacts with Kap60. The NLS major box of yRCC1 is unusual (17-RAKKMSK-23), lacking basic residues at both $P_3$ and $P_4$, but with R17 at position $P_{-1}$ that makes a strong bidentate contact with D276 (Supplementary Table 2). The electron density for the major NLS box is very strong (Fig. 6b), as suggested by the relatively low B-factor of refined atoms, lower than the average B-factor of the complex (~60 vs. 85 Å²). Third, a 'loop-helix' motif (residues 24–46) lies immediately C-terminal of the yRCC1 major NLS box and connects it to the first β-strand of the β-propeller (residues 47–51) (Fig. 7b). While the loop (residues 24–32) makes no contacts with Kap60, the short α-helix (residues 33–40) packs against helixes B of Arms 1–2, sandwiched between the yRCC1 β-propeller and the concave groove of Kap60 (Fig. 7a, c). This interaction is stabilized by 4 bonds with the notable contribution of two acidic residues in yRCC1, E34 and D35, that make salt bridges with Kap60 residues R112, Q113 (Fig. 7a, c) and H106. The α-helix in yRCC1 is not present in human RCC1, where the linker is 13 residues shorter than in yeast (Fig. 7b, Supplementary Fig. 6b). Finally, several residues scattered throughout the yRCC1 β-propeller (e.g., L41, V43, K166, K397, R333, and Q411) make additional contacts with Kap60 outside the NLS-binding grove (Fig. 7a).

To probe the importance of each of the four regions described above, we generated seven RCC1 constructs carrying mutations in the NLS and subjected them to pull-down analysis with GST-tagged Kap60 lacking the autoinhibitory IBB, as described for human RCC1. In this experiment, purified GST-ΔIBB-Kap60 and yRCC1 were incubated in solution at the same concentrations used for human factors (e.g., 1 µM ΔIBB-Kap60 and 0.75 µM yRCC1), followed by pull-down, SDS-PAGE analysis, and quantification (Fig. 7d, e). Strikingly, we found all point mutations at $P_2$ and $P_2'$ as well as the double $P_2/P_2'$ mutant completely disrupted yRCC1 association with Kap60, while mutations in the α-helix (E34/D35) did not affect binding to Kap60 (Fig. 7d, e). Likewise, deletion of the minor NLS-box alone (Δ10-yRCC1) or minor plus major NLS boxes (Δ23-yRCC1) completely obliterated association with Kap60 at the concentrations used in the assay (Fig. 7d, e). Thus, binding of yRCC1 to Kap60 is critically dependent on the minor NLS-box and requires a linker between the RCC1 NLS and β-propeller to position the latter domain onto the Arm-core.

## Discussion

This paper establishes the structural basis for nuclear import of RCC1, a cargo essential to generate a RanGTP gradient on chromatin[4]. Crystallographic and SAXS structures of human and yeast RCC1 in complex with importin α3 and Kap60 reveal the RCC1 NLS makes conserved contacts spanning the entire concave surface of importin α, while the β-propeller is recognized in different ways. It is flexibly wedged against the lateral surface of Arms 1–4 in humans ("backpack" conformation) (Fig. 8a), while sits onto Arms 1–2 of Kap60 in the yeast complex ("hat" conformation), rotated by ~60° with respect of human complex (Fig. 8b). This work not only sheds light on the structural evolution of RCC1 recognition by importin α but also provides clues to rationalize the diversification of importin α isoforms that took place in higher eukaryotes.

There are three factors that contribute to the increased complexity of the human RCC1 import signal compared to classical NLS sequences recognized by importin α1. First, RCC1 NLS is structurally and topologically complex. It lies in a flexible extended tail that samples multiple conformations in solution (Fig. 5) and is flanked by a bulky β-propeller domain folding back

onto the major-NLS box (Figs. 2a and 8a). Second, RCC1 NLS has multiple binding activities. It is involved in association with chromatin (both DNA and histones[39, 46]), importin α (Fig. 3b), and Ran[46] that allosterically regulates the RCC1 tail by binding to the β-propeller face opposite to importin α3 (Fig. 2a). Third, RCC1 NLS is post-translationally modified at multiple sites throughout the cell cycle[39, 42, 43], especially in the proximity of the minor NLS box (e.g., phosphorylation at S11, methylation at position 2–3) (Fig. 2a, b). Overall, RCC1 has poor structural complementarity for importin α1, explaining the low micromolar $K_D$ measured in vitro (Fig. 1b), insufficient to sustain nuclear import at physiological concentration of both factors[53].

A wealth of structural studies has shown the rigid[31] arch-like structure of importin α1 efficiently accommodates NLSs that are stretched in a helical groove by making mainchain and sidechains contacts with importin α1 sidechains[11–14]. This recognition is stereochemically possible and energetically favored for cargos that have a linear, bead-on-a-string topology such as the CAP80 complex[56], whereby the NLS is neighbored by discrete, non-interacting secondary structure elements (Fig. 8c). In contrast, the β-propeller of human RCC1 folds back onto the major NLS box, clashing with Arms 1–2 and preventing high-affinity association with importin α1 (Fig. 4b, Supplementary Fig 1B). Higher eukaryotes solved the RCC1 recognition problem by developing importin α3. This isoform is significantly more flexible than α1 and behaves in solution like a 'soft-spring'[31], stretching and extending akin to the receptor importin β[57, 58]. We previously identified a hinge-bending axis near Arms 3–4 of importin α3 that allows the first three repeats to 'flap' dynamically, opening up in a way other importin α isoforms are unlikely to do. The crystal structure of the importin α3:RCC1 complex presented in this paper (Fig. 2a) provides a direct structural validation of importin α3 built-in flexibility. The open conformation of Arms 1–3 accommodates the β-propeller domain of RCC1, while making high-affinity contacts with the NLS. This interaction is dynamic, as demonstrated by the eight, slightly different RCC1 conformers trapped in the triclinic cell (Supplementary Fig. 1a). Importin α3 plasticity and improved structural complementarity for RCC1 results in one order of magnitude higher binding affinity than importin α1 ($K_D = 0.62 \pm 0.1$ vs. $5.4 \pm 0.3$ μM) (Fig. 1), explaining the specificity observed in live cells[24–27] where the total concentration of importin α is ~1 μM[53]. Importin α3 selectivity for RCC1 is therefore a direct consequence of a flexible solenoid that undergoes local conformation changes at the N terminus to avoid steric clashes with the RCC1 β-propeller (Fig. 4b). Paradoxically, the amino acid sequence of the RCC1 NLS is not responsible for importin α3 selectivity, as demonstrated by replacing the β-propeller with GST or by introducing a 12-residue linker between the β-propeller and NLS that yield chimeras recognized equally well by importin α1 and α3 (Fig. 4d, e). Importin α3 selectivity for RCC1 is dictated by the ability of this isoform to accommodate the β-propeller rather than making more contacts with the NLS than other isoforms. Thus, the structural neighborhood where an NLS sequence lies rather than its primary sequence determines specificity for importin α3.

Other importin α3-specific cargos such as the influenza PB2 polymerase subunit and the NF-κB p50:p65 heterodimer bear topologically complex NLS sequences, also poorly compatible with importin α1. PB2 is part of a multisubunit polymerase complex that undergoes dramatic conformational changes to expose its NLS and become visible to the import machinery[31, 59]. Intriguingly, PB2 is also phosphorylated at S742[60], a residue that occupies a structurally equivalent position to S11 in RCC1 (Fig. 4c). It is remarkable that both PB2 and RCC1 show preferential binding to importin α3, make strong contacts at the minor NLS-binding site and are phosphorylated C-terminal of

this site. Likewise, the NF-κB p50:p65 heterodimer[29, 30] contains a complex NLS sequence formed at the heterodimeric interface of p65 and p50 and masked by the cytoplasmic inhibitor IκB under resting conditions. We propose the flexible solenoid of importin α3 has evolved to recognize import cargos that contain topologically complex NLS sequences, where considerable conformational rearrangement is required to unmask the NLS and recruit import factors. This perhaps explains why classical biochemical and structural studies with isolated NLS-peptides (so called "divide-and-conquer" approach) have failed to decipher the mechanisms of NLS-cargo specificity for importin α isoforms, which were identified nearly two decades ago[24].

Importin α isoforms have appeared throughout the evolution of eukaryotes, in parallel with the increasing complexity of higher organisms[21, 22]. An apparent conundrum is how baker's yeast achieves efficient nuclear import of RCC1 using the generic Kap60, without a dedicated isoform such as α3 (Supplementary Fig. 5). In this paper, we identified a "loop-helix" motif flanking the yRCC1 NLS that directly contacts Arms 1–2 of Kap60 (Fig. 7c). Mutations in this motif did not destabilize yRCC1 binding to Kap60 (Fig. 7d, e), suggesting it functions like a "spacer" rather than a binding determinant, used to position the β-propeller onto the N terminus of Kap60 and avoid steric clashes with the Arm-core (Fig. 8b, Supplementary Fig. 6a). Thus, the reduced genetic complexity of yeast, which has only one importin α gene vs. seven in humans, is compensated by a greater complexity of the RCC1 NLS. This suggests the RCC1 NLS and importin α co-evolved throughout evolution to minimize steric hindrance and maximize molecular recognition, essential to promote active nuclear import. While importin α specialized in multiple isoforms in higher eukaryotes, the RCC1 NLS became simpler (e.g., shorter) and likely more dependent on regulation by post-translational modification. Similarly, complex membrane protein NLSs promoting nuclear import of ER-synthesized membrane proteins appear to be more conserved in yeast than metazoa[15, 16], also pointing to a simplification of NLS sequences during evolution, accompanied by an expansion of trans-acting regulatory mechanisms of nuclear import.

Interestingly, three splicing isoforms of RCC1 exist in humans: the predominant isoform "α" that we crystallized with importin α3 (Fig. 2a) and two slightly longer, minor isoforms known as "β" and "γ". Putative loop-helix insertions, previously named NTR, for N-terminal regions[61], are also present in RCC1β and RCC1γ (Supplementary Fig. 6b). We speculate yRCC1-like insertions in RCC1β and γ function as extendable "spacers" that promote binding to importin α1 or α5 in those cell types that do not express significant quantities of importin α3. A more thorough and systematic analysis of importin α and RCC1 isoforms expression in human tissues is required to validate this hypothesis and shed light on nuclear transport of RCC1 minor splicing isoforms.

In conclusion, this paper defines molecular rules for nuclear import of human and yeast RCC1. We demonstrate that isoform α3 recognizes the three-dimensional context of human RCC1 rather than just its NLS amino-acid sequence. This work solves a long-standing problem in the biology of nuclear transport and paves the way to decipher how other importin α isoforms recognize NLS cargos vital to cell physiology (e.g., STATs) or associated with disease states (e.g., Ebola virus p24).

## Methods

**Molecular biology techniques**. A synthetic gene encoding human RCC1 (Supplementary Table 3) was synthesized by GenScript, while yeast RCC1 was PCR out from a *Saccharomyces cerevisiae* (strain S288C) genomic library. Both human and yeast RCC1 genes were cloned in vectors pET28a (Novagen) and pGEX-6P (GE Healthcare) between restriction sites Bam HI and Xho I. All deletion mutants

of human and yeast RCC1 (Δ5-RCC1, Δ10-RCC1, Δ10-yRCC1, Δ23-yRCC1, Δ42-yRCC1) were generated by PCR using loop-out primers. Site-directed mutagenesis was used to introduce point mutations in human RCC1 (S11E, R9A, K21A, R9A/K21A) and yeast RCC1 (R4A, K20A, R4A/K20A, E34A/D35A). GST-RCC1-long, constructed by insertion PCR, contains a 12-residue linker (GSAGSAAGS-GEF) between S26 and H27 of wild type (WT) GST-RCC1. GST-RCC1-NLS and GST-RCC1-NLS (P₂/P₂′) were constructed be introducing a stop codon at position 25 of the full-length GST-RCC1 plasmid. All constructs generated in this study were entirely sequenced to ensure the correctness of the DNA sequence. The nucleotide sequences of all primers used in this study is provided in Supplementary Table 3. Mouse importin α1[18], human importin α3,[31] and yeast Kap60[16] (with and without the IBB domain) were cloned in vectors pGEX-6P (GE Healthcare) and pET28a (Novagen). All plasmids were expressed in BL21(DE3) AI cells (Invitrogen) or regular BL21(DE3). To form stoichiometric complexes of RCC1 and importin α, plasmids encoding human GST-RCC1 and untagged ΔIBB-importin α3 or yeast GST-RCC1 and ΔIBB-Kap60 were co-expressed in BL21-AI cells (Invitrogen). Expression of recombinant proteins was induced with 0.5 mM IPTG (supplemented with 0.2% arabinose in BL21(DE3) AI cells), and grown overnight at 20°C. Cell pellets were suspended in Lysis buffer (20 mM Tris-HCl pH 8.0, 250 mM NaCl, 5 mM BME, 1 mM PMSF, and 0.2% Tween-20) and lysed by sonication. Clarified lysates expressing GST-tagged proteins were incubated with glutathione resin (GenScript) and beads were washed with Complex buffer (20 mM Tris-HCl pH 8.0, 150 mM NaCl, 1 mM EDTA, 5 mM BME, 0.1 mM PMSF). Clarified lysates expressing His-tagged proteins were incubated with Ni-agarose resin (GenScript) and beads were washed with High Salt buffer (20 mM Tris-HCl pH 8.0, 600 mM NaCl, 3 mM BME, 0.1 mM PMSF) and eluted with Imidazole Elution buffer (20 mM Tris-HCl pH 8.0, 150 mM NaCl, 100 mM Imidazole, 3 mM BME). The RCC1:importin α and yRCC1:Kap60 complexes were cleaved off beads by overnight digestion with PreScission Protease and further purified by size exclusion chromatography on a Superdex 200 16/60 (GE Healthcare) pre-equilibrated in Complex buffer. All species were concentrated by ultra-filtration using Millipore concentrators.

**Isothermal titration calorimetry.** ITC experiments were carried out at 20 °C using a nano-ITC calorimeter (TA Instruments). Prior to ITC analysis, both GST-RCC1 and ΔIBB-importin αs were dialyzed overnight against Gel Filtration (GF) buffer (20 mM Tris-HCl pH 8.0, 150 mM NaCl, 5 mM BME, 0.1 mM PMSF) at 4 °C. GST-RCC1 (300 μM) was injected in 2.0 μl increments into a calorimetric cell containing 195 μl of ΔIBB-importin α3 or α1 (100 μM). The spacing between injections was 180 s. Titrations were performed in triplicate and data were analyzed using the NanoAnalyze data analysis software (TA Instruments). Heats of dilution were determined from control experiments carried out by injecting GF buffer against importin α3/α1 and subtracted from enthalpies obtained by titrating RCC1 against importin α3/α1. Curve fitting was done in NanoAnalyze data analysis software using a single set of binding sites model. The concentration of samples used for ITC was accurately determined using amino acid analysis, Lowry assay and spectrophotometric determination with the theoretical extinction coefficient ε. In both titrations in Fig. 1, the N-value at the midpoint point is <1, as expected for a 1:1 interaction, because the active fraction of recombinant importin α isoforms in cell is likely <1.

**Pull-down assay.** Pull-down assays in Figs. 3b and 7d were carried out by pre-incubating physiological concentrations of purified GST-ΔIBB-Importin α/GST-ΔIBB-Kap60 (1 μM) and RCC1/yRCC1 (0.75 μM)[53] for ~60 min at 4 °C. A volume of 700 μl of this mixture was then incubated with 100 μl of glutathione resin beads (GenScript) pre-packaged in a mini spin-column, for 1 h at 4 °C. Beads were then washed three times with 700 μl of complex buffer and GST-Importin α/Kap60:RCC1/yRCC1 complexes were eluted with 100 μl of complex buffer containing 25 mM reduced glutathione. 20 μl of this elution was then analyzed by SDS-PAGE (15%): gels were stained with Coomassie Brilliant Blue-G-250, destained overnight and quantified. An uncropped image of the gels shown in Figs. 3b and 7b is provided in Supplementary Fig. 8. Pull-down experiments to determine the specificity of importin α1/α3 for RCC1 (Fig. 4d) were carried out by pre-incubating physiological concentrations of GST-RCC1, GST-RCC1-long, GST-RCC1-P2/P2′, GST-RCC1-NLS (residues 1–24), GST-RCC1-NLS (P₂/P₂′) with His-tagged ΔIBB-importin α1 or α3, as described above. For quantification of pull-downs in Figs. 3b and 7d, the intensity of WT RCC1/yRCC1 bands pulled-down by GST-ΔIBB-importin α3/GST-ΔIBB-Kap60 was normalized to 100% and pixel intensities of bands corresponding to RCC1 mutants were compared to the WT protein. Conversely, for quantification of pull-downs in Fig. 4d, the pixel intensity of ΔIBB-importin α3 pulled down by GST-RCC1 was normalized to 100% and pixel intensities of bands corresponding to ΔIBB-importin α3 and α1 pulled down by GST-RCC1 mutants and GST-RCC1-NLS constructs were compared accordingly.

**Crystallographic methods.** ΔIBB-importin α3:RCC1 and ΔIBB-Kap60:yRCC1 were crystallized by mixing equal volumes of protein complex at 20 mg ml⁻¹ with crystallization solutions containing 0.1 M sodium cacodylate buffer (pH 6.5), 0.2 M calcium, 8% PEG 8000, galactose 3% and 0.1 M Na cacodylate, 0.1 M NaCl and 1.1

M ammonium sulfate, respectively. Crystals were harvested in nylon cryo-loops, cryo-protected with 27% ethylene glycol and flash-frozen in liquid nitrogen. Crystals were diffracted at LS-CAT Beamline 21-ID-F at Argonne Photon Source on a MARMOSAIC 225 CCD detector and at beamline 14-1 at SSRL on a Rayonix MX325 CCD detector. Data indexing, integration, and scaling were carried out with HKL2000[62]. Initial phases were obtained by molecular replacement using Phaser[61] and PDB 3Q5U and 4PVZ as search models. Atomic models were built using Coot[63] and refined using phenix.refine[64]. Final stereochemistry was validated using PROCHECK[65]. Data collection and refinement statistics are summarized in Table 1.

**SAXS analysis and ab initio shape reconstruction.** The monodispersity of all samples used for SAXS analysis was validated using AUC-SV and SEC (Supplementary Fig. 7, Supplementary Table 1). SAXS data for the importin α3:RCC1 complex and human RCC1 were measured using a Rigaku BioSAXS-2000 instrument at the Sidney Kimmel Cancer Center X-ray Crystallography and Molecular Interaction Facility, at Thomas Jefferson University. A volume of 30 μl of sample was pipetted into 1.0-mm quartz capillary cells that were sealed at each end with screw caps and O-rings for measurement under vacuum. All samples were dissolved in GF buffer at concentrations between 1.0 and 7.5 mg ml⁻¹. Samples were centrifuged at 16,000 × g for 10 min prior to exposing to X-rays, which was done for 5–30 min in triplicates. The scattering of GF buffer alone was subtracted from all protein sample scattering data. Guinier analysis between the triplicate exposures was used to attentively control for radiation damage and protein aggregation. Data reduction was carried out by circular averaging of the images and scaling to obtain the scattering curve (scattering intensity ($I$) as a function of the momentum transfer vector $q$ ($q = 4\pi(\sin\theta)/\lambda$) using the Rigaku Automatic Data Analysis Pipeline software. The program GNOM[66] was used to calculate $P(r)$ plots from scattering data. SEC-SAXS data for the Kap60:yRCC1 complex were collected at G1 station at MacCHESS, which usually operates at 9.8 keV and has a 1.5 m camera length and typically reaches q_min = 0.007 to q_max = 0.3. SEC was performed using an AKTA FPLC system equipped with a Superose 12 10/300 GL column (GE Healthcare). Data were recorded on a Pilatus 100K-S detector. The Kap60:yRCC1 complex at 5.0 mg ml⁻¹ was loaded onto the size exclusion column previously equilibrated in a buffer containing 20 mM Tris-HCl pH 7.5, 150 mM NaCl, 5% glycerol and 3 mM BME. Primary reduction of the SAXS data was performed using RAW[67], and ATSAS software[68]. Frames 710–750 were used to calculate experimental scattering intensities. Ab initio model calculations to generate a scattering envelope were done using DAMMIF[69]. Twenty solutions obtained from DAMMIF were used to check consistency and averaged together to obtain the final model using the DAMAVER program suite[70]. This final averaged model was then converted to a surface map using the SITUS program suite[71]. Theoretical solution scattering curves of all crystallographic structures determined in this work were calculated using the FoXS web server[72]. SAXS data collection and analysis statistics are summarized in Supplementary Table 1.

**SEC and AUC.** Analytical SEC was carried out on a Superpose 12-column equilibrated with Gel Filtration buffer (20 mM Tris-HCl pH 8.0, 150 mM NaCl, 5 mM BME, 0.1 mM PMSF). AUC-SV analysis was carried out in a Beckman XL-A Analytical Ultracentrifuge out at the Sidney Kimmel Cancer Center X-ray Crystallography and Molecular Interaction Facility at Thomas Jefferson University. Importin α3:RCC1 complex, human RCC1 and Kap60:yRCC1 complex were dissolved at 0.8–2.5 mg ml⁻¹ in 20 mM Tris-HCl pH 7.5, 150 mM NaCl, 5% glycerol, 3 mM BME and were spun at 201,600 × g (equal to 50,000 r.p.m. in the AN-50 Ti analytical rotor) at 10 °C. Absorbance values at 280 nm were fit to a continuous sedimentation coefficient ($c(s)$) distribution model in SEDFIT[73].

**Structure analysis.** Molecular interactions were analyzed using PISA[74] and binding energies between RCC1 and importin α/Kap60 were predicted using PRODIGY[75]. All figures in the paper were prepared using the program PyMol[76].

**Data availability.** Coordinates and structure factors for the importin α3:RCC1 and Kap60:yRCC1 complexes have been deposited in the protein Data Bank (accession codes 5TBK and 5T94). Other data are available from the corresponding author upon reasonable request.

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

## Acknowledgements

We are thankful to the staff at APS beamline 21-ID-F and SSRL station 9:2 for beamtime and assistance in data collection. We thank the staff at macCHESS for help with SEC-SAXS data collection. R.A.P. was supported by NIH grant T32 GM100836. This work was supported in part by NIH grants GM074846, GM100888, and CA56036. Research in this publication includes work carried out at the Kimmel Cancer Center X-ray Crystallography and Molecular Interaction Facility at Thomas Jefferson University, which is supported in part by NCI Cancer Center Support Grant P30 CA56036 and S10 OD017987. CHESS is supported by the NSF award DMR-1332208, and the MacCHESS resource is supported by NIGMS award GM-103485.

## Author contributions

R.A.P. obtained initial crystals of the RCC1:importin α3 complex. R.S.S. and R.K.L. perfected crystallization, obtained crystals of the yRCC1:Kap60 complex and determined all crystallographic structures and carried out SAXS with the help of G.C. R.E.G. and R.S.S. carried out collection and analysis of SEC-SAXS data. R.S.S., R.K.L., and S.B. generated mutations, purified complexes, and carried out pull-down assays. G.C. supervised the entire project and wrote the paper with help from all authors.

## Additional information

**Competing interests:** The authors declare no competing financial interests.

