## [Peer Review file · Nature Communications]

Reviewers' comments:

Reviewer #1 (Remarks to the Author):

One of the main questions in the nuclear transport field is to understand cargo specificity. What dictates that one protein binds to a specific nuclear transport receptors(NTRs), while another cargo binds to another NTR? Why do humans have so many different NTRs when yeast can live with much fewer?

In their study the authors address this problem by studying the interaction of RCC1 with Imp-a1 and Imp-a3, respectively. RCC1 has long been known to preferentially bind Imp-a3 over other importins, but a structural basis is missing. Two complex crystal structures are presented – human RCC1 bound by Imp-a3, and yeast RCC1 bound to Kap60, the yeast Imp-a homolog. Paired with strong biochemical data, the authors make a good case for why Imp-a3, compared to Imp-a1, has a 10fold higher affinity for RCC1. In essence, the authors argue that the propeller domain of RCC1 abuts Imp-a3 in a way that is likely to be sterically unfavorable in other importins, hence reducing affinity. The structural data is supported by pull-down data.

The structural work is carried out very well and leaves no room for criticism.

I have a number of smaller concerns that I would like to see addressed in a revised manuscript.

These concerns are partially technical, but primarily about data presentation.

- 1.) Figure 1: x-axis should read 'Molar Ratio'. More importantly, why is the transition midpoint at ~0.25-0.3? If we are looking at a 1:1 reaction I would expect the transition midpoint to be at 1. This needs to be explained. Could it be that either the protein concentrations are off or that one of the two proteins is only partially functional or partially pure, thereby reducing its effective concentration?
- 2.) Figure 2: Panel a should be left, panel b to the right, to follow standard convention. Highlight the aa23-27 hinge area in Imp-a3.
- 3.) One big point of the paper is the interaction of the RCC1 propeller with Imp-a3 which would lead to clashes in Imp-a1. Judged by Suppl. Fig1 and Figure 2d, the relative position of the propeller domain with respect to Imp-a3 changes quite significantly between the eight copies of asymmetric unit. Thus, it is not immediately obvious how severe those clashes in Imp-a1 would really be, given the flexibility around hinge aa23-27 of RCC1. In addition, it should be discussed whether crystal packing influences the propeller position, since this is highly relevant to the Imp-a1 clash argument. Instead of Figure 4b, it would be more convincing to show the two most divergent RCC1-propeller positions found in the crystal and superpose each on Imp-a1 in cartoon display (not solid cylinders, since they can be distorted)
- 4.) Extending on the theme of the importance of the propeller domain for Imp-a3 preference, the authors have a great pull-down assay in their hands (Fig 4c). Rather than clipping the propeller off entirely, I wonder whether the author could do an experiment where the hinge between NLS and propeller is extended and/or made more flexible. I think this could proof the point that steric considerations affect the Imp-a3 preference through negative selection of Imp-a1, rather than additional contacts outside the NLS that positively select for Imp-a3.
- 5.) Figure 5 could go in the supplement.
- 6.) The affinity to Imp-a1 is recorded at 5.4uM, but in the text it is described as 'high micromolar binding affinity'. Please adjust wording to 'low'.
- 7.) When describing the van der Waals interactions (lines 134 and 265) it would be better to mention the number of residues involved, rather than the number of contacts.
- 8.) Line 99 – the structural comparison to Nup85-Seh1 of the Y complex seems out of place and should be taken out, including references 48 and 49.

Reviewer #2 (Remarks to the Author):

Sankhala and co-authors address a lingering question in nucleocytoplasmic transport. Specifically, how do different iso-forms of Karyopherin- α , of which there are seven in humans, mediate selective cargo binding despite sharing, invariant, heavily conserved binding sites. Towards this goal the authors present two crystal structures of yeast and human RCC1 in complex with the universal yeast karyopherin- α Kap60 and the specialized human Karyopherin- α 3, respectively. These structures, particularly the human RCC1:Kap- α 3, are technically impressive and represent a significant advance in the study of these versatile transport molecules. Furthermore, the structural work is carefully validated and expanded upon by extensive biochemical analysis, including SAXS and ITC experiments to establish stoichiometry and binding constants. RCC1 is a critical Ran guanine nucleotide exchange factor, which is localized in the nucleus, where it maintains a RanGTP gradient responsible for driving a broad range of essential cellular pathways, including nucleocytoplasmic transport. Elucidating its novel attachment to the nucleocytoplasmic transport machinery represents a standout advance on its own terms. However, this study also allows the conclusion that specialized Karyopherin- α iso-forms have evolved to accommodate complex topographical binding interfaces, remote of the heavily studied NLS motifs, supplying a selective pressure for these alternative transport adaptors. This conclusion is an exciting advance within the nucleocytoplasmic transport field and provides an invaluable prototypical example of how unconventional nuclear cargo recognition occurs. The presented findings are of broad interest to the readership of Nature Communications and undoubtedly of substantial impact within the nucleocytoplasmic transport field. Consequently, the manuscript should be accepted for publication. The author may wish to consider the points below which in my opinion will further elevate the quality and accessibility of the manuscript.

Major point

(1) In several figures quantitative pull-downs are employed to demonstrate relative binding affinities. However, it is customary to present also input SDS-PAGE gels to demonstrate that the proteins were loaded in equal amounts. However, these SDS-PAGE gels are missing for Figures 3B, 4C, 7D and S3 and should be provided in the revised manuscript.

Minor points to figures

While the manuscript is easy to follow, in my opinion the figures require the attention of the senior author to augment both the aesthetics and accessibility of the otherwise beautiful results. I would like the authors to consider the following points.

(1) Throughout the manuscript the figure resolutions are uniformly low, particular examples include Figures 2, 5, S1, S2 and S4. I cannot judge whether this is the result of the Nature Communications manuscript handling system or whether these figures were of low resolution to begin with. Regardless, in the published manuscript the figure resolution should be increased.

(2) There are broad alignment issues with panels and especially panel labels placed unevenly across multiple figures. Fig 6 is an example of this.

(3) Figures 2A and 2B show the peptide in a different shade of green. Furthermore, the electron density is not of figure quality. I would like to suggest a consistent color scheme for the NLS using the same darker shade as in Figure 2A, and adjusting the sampling rate and mesh width of the map in pymol to produce a less coarse electron density map. Moreover, I would like to ask the authors to also consider unifying the σ level used throughout the manuscript to be directly comparable with Figure 6B.

(4) While I appreciate the principle behind Figure 3A and 7A, these schematic are unclear and

would benefit from a re-work. Perhaps a simplification of the cartoon aspects by increasing the spacings between residues and changing the cartoon style used for the side-chains is all that is needed to increase clarity. However, please ensure to keep the segregation between distinct binding surfaces, this idea is excellent!

(5) The two shades of grey used for the Kap- α 3 chains in Figure 4A are difficult to distinguish and I would suggest to employ more distinct colors. The SAXS envelopes in Figures 5B and 6D, as well as the γ RCC1 surface in Figure 7C are too transparent. Additionally, the SAXS envelope color should match Figure 2D.

(6) In Figure 4A an additional panel highlighting the local environment of the modified S11 and S742 residues shown in ball-and-stick representation would be an important addition.

(7) In Figures 2C, 5A, and 6C, the insert graphs for the SAXS experiments should be presented as separate panels for increased clarity.

(8) In Figure 8, a superposition of the two RCC1 structures aligned on the karyopherins, displaying the differing orientations and packing of RCC1, would provide more information than the current side-by-side representation.

(9) In Figure S3, the molecular weight ladder needs to be labeled.

Minor points to manuscript text

In general the manuscript is clearly written, concise and easy to follow. However, there are a few suggestions I would like the authors to consider.

(1) Page 3 lines 71-73. A consistent approach to addressing RCC1 modifications would be appreciated, annotate positions 2-3 within the text with 3 letter residue codes.

(2) Page 4 Lines 95-97. This is a confusing sentence! I would suggest to split up the sentence into two sentences: a full sentence discussing the cell contents and the large degrees of flexibility observed, and then address how this relates to the challenge of refining this low-resolution structure and the respectable refinement statistics achieved. Also, the range of conformations shown in Figure S1 in main text Figure 5 should be discussed.

(3) Page 15 onwards refers to a 'Complex buffer' which is subsequently termed 'Gel filtration' buffer. Consider consolidating the nomenclature to a single name for each unique buffer used.

(4) The reported crystal structures, in light of their lower resolution, would benefit from additional validation using Mol-Probity. I would suggest to include the Mol-Probity score and refinement clash scores in Table S1.

Reviewer #3 (Remarks to the Author):

General comments:

The authors applied small angle scattering to three different systems: RCC1-importin complex, RCC1 and RCC1-Kap60 complex. While all three systems have some problematic parts (outlined in detail below), the conclusions drawn for RCC1-importin complex still seem robust and correct. For RCC1, the quality of the original data raises complications, which should be addressed in the manuscript. Nevertheless, the interpretation of the data seems acceptable. For RCC1-Kap60 complex, the quality of the original data is dubious. For this system, a SEC-SAXS experiment is most likely necessary to avoid interpreting artifacts. As it is, the data from RCC1-Kap60 is not suited for publication.

In general, the authors' representation of the SAXS data is rather unconventional. I would recommend following the guidelines of the IUCr for reporting BioSAXS data, which can be found here: <http://journals.iucr.org/d/issues/2012/06/00/be5200/index.html> The authors use the term "gyration radius (Gr)" in place of the considerably more common "radius of gyration (RG)". Referring to the high resolution, crystallographic models as X-ray models is not good nomenclature, as the ab initio models derived from the SAXS data are also "X-ray" models. Crystallographic models would be a more suitable term.

P 5: SAXS analysis of the RCC1-importin complex

The authors state that several concentrations were measured and that further analysis was performed for all concentrations, but do not describe how robust their findings were and which data was used for the model fitting and figures. It should be noted that performing individual analysis at each concentration, instead of performing extrapolation to zero concentration, is a very unusual approach.

In addition, the concentrations reported in the text and those in figure S2 do not match.

While the χ^2 of the FOXS fit is indeed good, the figure 2c shows some deviations. The term "excellent" is therefore a bit exaggerated. In particular, the fact that both RG and D_{max} of the predicted curves are smaller than observed is a better indicator for the postulated elongation than the comparison of ab initio SAXS models and crystallographic models.

Given these deviations, they authors should consider using more advanced modeling tools such as sreflex from the ATSAS package to confirm that an elongation would result in a match between predicted and measured SAXS data.

The comparison between the predicted and the measured SAXS curves shows some significant deviations, both in $I(q)$ vs. q and the $p(r)$ functions. Judging from the $p(r)$ function this might be due to a small, but not negligible presence of higher oligomers.

Despite these issues, based on the data shown, the conclusion that the crystallographic model and the SAXS data agree well seems correct.

P8. SAXS analysis of RCC1

As previously, it is not clear how the protein concentration affects the results of the analysis.

Again, the concentrations reported in the text and those in figure S2 do not match.

As figure S2 B clearly shows a concentration dependence of RG, the statement that the data is consistent with a monomer cannot be generally true.

Figure 5, in particular the $P(r)$ function, clearly shows larger objects than the crystallographic model contributing to the SAXS data. In addition with the concentration dependence of the radius of gyration, this raises the question to which extent the racket handle is an artifact due to higher oligomers or aggregation. This caveat of the interpretation should be mentioned in the manuscript.

P9: SAXS analysis of RCC1-Kap60 complex

As for the previous two samples, it is not clear how the protein concentration affects the results of the analysis, and the concentrations and radii of gyrations reported in the text and those in figure S2 do not match. In particular the statement that the samples are monodisperse below 2.5 mg/mL

seems dubious, as the SAXS signal apparently is not concentration independent in this range. The statement that "An ab initio shape reconstruction from scattering data at 1, 2 and 2.5 mg ml⁻¹ reveals an elongated shape (Fig. 6d), similar to a boomerang, in reasonable agreement with the X-ray structure ($\chi^2 \sim 3.0$)" seems to imply that the χ^2 of the ab initio model is 3.0, which seems unlikely. It should be clearly stated that the SAXS data was compared to predicted curves from a crystallographic model.

As a matter of fact, the predicted and measured SAXS do not match at all. As the SAXS data seems to have some aggregates from higher oligomers/aggregates, this is not surprising. RCC1-Kap60 probably needs to be measured with SEC-SAXS to address the issue of sample monodispersity.

P16: Methods of SAXS analysis and ab initio shape reconstruction

line 486: The composition of the gel filtration buffer is never explicitly stated. Is it the same as the complex buffer described on page 15, line 423?

Line 471 - 474: Which software was used for data reduction and Guinier analysis? It also seems very probable that data reduction was performed prior to the Guinier analysis.

Line 479: I would assume that FoXS was used to calculate all theoretical scattering curves in the manuscript, not only the one of importin $\alpha 3$ -RCC1 as stated.

Line 480: Instead of supplementary table 1, supplementary table 2 should be referenced here. However, table 2 contains no data collection statistics.

This section (nor any other part of the manuscript) contains now reference whatsoever on how the monodispersity of the SAXS data was assessed. As the interpretation of the data depends strongly on the monodispersity, this is a major flaw.

P. 22: Caption for Figure 2 & P23: Captions for Figure 5 & 6

All SAXS data, including those in panel c) should be clearly identified (concentration!)

Figure 2c, 5a and 6c:

The y-axis label on the $I(q)$ vs. q plot should be $\log(I(q))$ not $I(q)$

Figure S2:

The presented concentrations do not match with those presented in the relevant sections of the main body of the manuscript.

This figure is the only place in the manuscript where the authors use the abbreviation RG for the radius of gyration.

The radius of gyration in the RG vs concentration plots requires units.

The Kratky plots should be presented normalized for RG. Also, it should be stated to which concentration they belong.

Guinier plots for all presented SAXS data should be added to this figure.

Supplementary table 2:

Contrary to its caption the table contains no information on the SAXS data collection.

Point-by-point Response to Reviewers for paper NCOMMS-17-03350

Reviewers' critiques were received on April 11th, 2017. We thank the reviewers for taking their precious time to read our manuscript and for making useful comments that improve the quality of our work. Below we address each of the specific points raised by the three reviewers (all revisions in the paper are colored in red).

REVIEWER 1

1. Figure 1: x-axis should read 'Molar Ratio'. We changed 'Mole Ratio' to 'Molar Ratio'. Sorry, this was a typo. More importantly, why is the transition midpoint at ~0.25-0.3? If we are looking at a 1:1 reaction I would expect the transition midpoint to be at 1. This needs to be explained. Could it be that either the protein concentrations are off or that one of the two proteins is only partially functional or partially pure, thereby reducing its effective concentration? The transition point is below 1 because the **Active Fraction** of purified importin α s is less than 1. The N-value measured by ITC is equal to the stoichiometry of binding only if: (i) the concentrations of reactants in cell and syringe used for fitting are 100% correct; (ii) the species in cell and syringe are 100% active (e.g. Active Fraction = 1). The N-value should be more accurately expressed as:

$N = \text{Stoichiometry} \times [\text{Active Fraction Cell}] / [\text{Active Fraction in Syringe}]$ that for Fig. 1 is:

$N = 1 \times [\text{Active Fraction of } 100 \mu\text{M importin } \alpha 1 \text{ or } \alpha 3] / [\text{Active Fraction of } 300 \mu\text{M RCC1}]$.

In our experiment, we routinely estimate the **concentrations** of all samples used for ITC using three independent methods (e.g. amino acid analysis, Lowry assay and spectrophotometric determinations using the theoretical extinction coefficient ϵ), which yield consistent values within the expected error range in biochemistry (~5-10%). However, we have limited control over the **Active Fraction** of recombinant importins. In our experience, there is always a significant portion of recombinant importin α that is not active, even if all our recombinant proteins are purified under native conditions, in the absence of detergents. This possibly explains why we use concentrations of importin α one to two orders of magnitude above the observed K_d (e.g. 100 μM in cell for K_d s ~ 0.5 to 6 μM). Fortuitously, the Active Fraction of importin $\alpha 1$ and $\alpha 3$ in the two reactions in **Fig. 1a** and **1b** is very similar suggesting a comparable amount of inactive importin α exists in both experiments. Nonetheless, the heat released upon injection of RCC1 against importin $\alpha 3$ (**Fig. 1a**) is undoubtedly much greater than that released with importin $\alpha 1$ (**Fig. 1b**), consistent with a tighter binding association (which was also validated by our pull-down assay (**Fig. 4d,e**)). In summary, to clarify this point, we added the following statement at page 16 (line 459) of the revised paper: ***"The concentration of samples used for ITC was accurately determined using amino acid analysis, Lowry assay and spectrophotometric determination using the theoretical extinction coefficient ϵ . In both titrations in Fig 1, the N-value at the midpoint point is less than 1, as expected for a 1:1 interaction, because the active fraction of recombinant importin α isoforms in cell is likely less than 1"***.

2. Figure 2: Panel a should be left, panel b to the right, to follow standard convention. Highlight the aa23-27 hinge area in Imp-a3. **Done!** RCC1 hinge residues (23-27) are now colored in red.

3. One big point of the paper is the interaction of the RCC1 propeller with Imp-a3 which would lead to clashes in Imp-a1. Judged by Sup. Fig1 and Figure 2d, the relative position of the propeller domain with respect to Imp-a3 changes quite significantly between the eight copies of asymmetric unit. Thus, it is not immediately obvious how severe those clashes in Imp-a1 would really be, given the flexibility around hinge aa23-27 of RCC1. To clarify this point, we superimposed importin $\alpha 1$ onto the structure of importin $\alpha 3$ bound to RCC1 most distant conformers (referred to as conformer #1 and #2 in **Fig. R1**), and zoomed in at the boundary between the β -propeller and Arm 1-2. In both cases, steric clashes occur between the two molecules, supporting our hypothesis that importin $\alpha 3$ evolved to accommodate

the β -propeller. To quantify the degree of “steric incompatibility” between importin α 1 and RCC1 conformers, we computed a “Clashscore” (as defined by MolProbity (1)), which ranges between **29.8 - 35.2** for α 1 bound to RCC1 most distant conformers, but is about 11 for importin α 3.

In addition, it should be discussed whether crystal packing influences the propeller position, since this is highly relevant to the Imp- α 1 clash argument. The importin α 3:RCC1 complex crystallizes in a triclinic space group with eight complexes in the unit cell. The eight complexes are in different positions of the P1 cell and adopt non identical crystal contacts. There is no obvious correlation between crystal contacts and RMSD of RCC1 conformers. To clarify these concepts we added the following sentence at page 4 (line 113) of the revised paper: “...**although there is no obvious correlation between crystal contacts and the RMSD of RCC1 protomers crystallized in the P1 cell.**” Instead of Figure 4b, it would be more convincing to show the two most divergent RCC1-propeller positions found in the crystal and superpose each on Imp- α 1 in cartoon display (not solid cylinders, since they can be distorted). We have tried dozens of different ways to illustrate the clashing between importin α 1 and RCCA β -propeller and found the cylinder representation in Fig 4b combined with a bead-on-a-string representation of importin α 1/ α 3 in panel 4a is the most intuitive way to describe this concept. To satisfy the reviewer curiosity, we added Fig. R1 to the revised paper as Suppl. Figure 1b.

4. Extending on the theme of the importance of the propeller domain for Imp- α 3 preference, the authors have a great pull-down assay in their hands (Fig 4c). Rather than clipping the propeller off entirely, I wonder whether the author could do an experiment where the hinge between NLS and propeller is extended and/or made more flexible. I think this could proof the point that steric considerations affect the Imp- α 3 preference through negative selection of Imp- α 1, rather than additional contacts outside the NLS that positively select for Imp- α 3.

To address the reviewer’s critique, we generated a new construct of RCC1 containing a 12 aa extension linker (GSAGSAAGSGEF) between residues S26 and H27, which we refer to as Long-RCC1. We repeated the pull-down assay in Fig. 4d by comparing the binding of importin α 1 and α 3 to WT RCC1, RCC1-Long and, as negative control, RCC1-P₂/P₂’ (Fig. R2). This experiment revealed that while RCC1 binds specifically to importin α 3, RCC1-Long loses specificity for this isoform and binds equally well to α 1 and α 3. As expected, the negative control RCC1-P₂/P₂’ has no detectable binding to either isoform. As previously shown, a smaller construct lacking the β -propeller also loses specificity for importin α 3 and associates better with importin α 1. Thus, this experiment confirms our model that importin α 3 selectivity for RCC1 is dictated by this isoform ability to accommodate the β -propeller rather than making additional contacts with the NLS as compared to importin α 1. We made the following changes to the revised paper: (i) we included Fig. R2 as Fig. 4d-e. (ii) at page 2 (line 32), we added: “...or inserting a 12 residue linker between NLS and β -propeller...”; (iii) at page 7 (line 199), we included this sentence: “**In contrast, a slightly longer construct of RCC1 (RCC1-Long) that bears a 12 residue linker (GSAGSAAGSGEF) between residues S26 and H27, right at the junction between the β -propeller and NLS, efficiently bound both importin α 1 and α 3, showing no selectivity for the isoform 3 (Fig. 4d,e).**” (iv) We slightly revised the Discussions at page 13.

Fig. R2: Pull-down assay probing the specificity of RCC1 (or just its NLS) for importin α 1 and α 3.

5. Figure 5 could go in the supplement. We believe Fig. 5 conveys an important concept and we’d like to keep it in the main text. Previous papers describing the structure of the apo-RCC1 (2), or RCC1 bound to the nucleosome core particle (3), described this GEF as a ‘globular’ protein. In light of the SAXS analysis presented in Fig. 5, we believe this is an inaccurate representation of RCC1. This protein has an extended N-terminal tail that spans ~50 Å and thus represents a prominent feature in solution.

6. The affinity to Imp- α 1 is recorded at 5.4 μ M, but in the text it is described as ‘high micromolar binding affinity’. Please adjust wording to ‘low’. **Done!** We changed ‘high’ to ‘low’ (line 338).
7. When describing the van der Waals interactions (lines 134 and 265) it would be better to mention the number of residues involved, rather than the number of contacts. **Done!** At page 5 (line 139) we revised the text: “...**23 residues of RCC1 interact with 39 residues of importin α 3, which account for...**”. At page 10 (line 277) we added: “**This binding interface is stabilized by 27 residues in γ RCC1 that interact with 47 residues of Kap60...**”
8. Line 99 – the structural comparison to Nup85-Seh1 of the Y complex seems out of place and should be taken out, including references 48 and 49. **Done!** We removed the sentence and Ref 48-49

REVIEW 2

Major Points

1. In several figures quantitative pull-downs are employed to demonstrate relative binding affinities. However, it is customary to present also input SDS-PAGE gels to demonstrate that the proteins were loaded in equal amounts. However, these SDS-PAGE gels are missing for Figures 3B, 4C, 7D and S3 and should be provided in the revised manuscript. We apologize for not explaining this point better. In **Suppl. Figure 3**, we provide loading controls for all proteins and pull-downs shown in **Figs 3b, 4d, 7d**. Given the large number of samples, it would have been difficult to include these loading controls in the main text. **Suppl. Figure 3** also confirms all proteins used in our pull-downs are genuinely pure. **Suppl. Figure 3** is now referenced in the legend of **Figs. 3b, 4d, 7d**.

Minor Points

1. Throughout the manuscript the figure resolutions are uniformly low, particular examples include Figures 2, 5, S1, S2 and S4. I cannot judge whether this is the result of the Nature Communications manuscript handling system or whether these figures were of low resolution to begin with. Regardless, in the published manuscript the figure resolution should be increased. We apologize for the poor quality of our images. The conversion to pdf during submission killed the quality of our illustrations. We uploaded beautiful (600 ppi) images that will be used for the final accepted manuscript.
2. There are broad alignment issues with panels and especially panel labels placed unevenly across multiple figures. Fig 6 is an example of this. **Done!** We have fixed the alignment of panels in all figures.
3. Figures 2A and 2B show the peptide in a different shade of green. Furthermore, the electron density is not of figure quality. I would like to suggest a consistent color scheme for the NLS using the same darker shade as in Figure 2A, and adjusting the sampling rate and mesh width of the map in pymol to produce a less coarse electron density map. Moreover, I would like to ask the authors to also consider unifying the σ level used throughout the manuscript to be directly comparable with Figure 6B. **Done!** We standardized the shade of green in **Fig 2a-b**. As far as displaying electron densities, the signal level was already standardized (though displaying difference maps). The 8-fold ncs-averaged 2Fo-Fc map in **Fig. 2b** is displayed at 1σ above background, while the Fo-Fc map in **Fig. 6b** is displayed at 2σ above background
4. While I appreciate the principle behind Figure 3A and 7A, these schematic are unclear and would benefit from a re-work. Perhaps a simplification of the cartoon aspects by increasing the spacings between residues and changing the cartoon style used for the side-chains is all that is needed to increase clarity. However, please ensure to keep the segregation between distinct binding surfaces, this idea is excellent! **Done!** We have simplified the schematic diagram in **Fig. 3a** and **7a** that we hope the reviewer will find easier to read.
5. The two shades of grey used for the Kap- α 3 chains in Figure 4A are difficult to distinguish and I would suggest to employ more distinct colors. The SAXS envelopes in Figures 5B and 6D, as well as the γ RCC1 surface in Figure 7C are too transparent. Additionally, the SAXS envelope color should match Figure 2D. **Done!** We updated **Fig. 4a** with two different colors for importin α 3 conformations (dark gray for α 3:RCC1 and

orange for $\alpha 3$:PB2). Likewise, we standardized the color and transparencies of all SAXS envelopes in **Figs. 2d, 5b, 6d** as well as we reduced the transparency of **Fig. 7c** by 40%.

6. In Figure 4A an additional panel highlighting the local environment of the modified S11 and S742 residues shown in ball-and-stick representation would be an important addition. **Done!** We added a second zoom-in panel (**Fig. 4c**) describing the local environment of S11 and S742. We also updated the figure legend at page 24 (line 742): “**Zoom-in panel showing the local environment of the RCC1 S11 and PB2 S742, which are both subjected to phosphorylation**”.

7. In Figures 2C, 5A, and 6C, the insert graphs for the SAXS experiments should be presented as separate panels for increased clarity. **Done!**

8. In Figure 8, a superposition of the two RCC1 structures aligned on the karyopherins, displaying the differing orientations and packing of RCC1, would provide more information than the current side-by-side representation. **Done!** A new figure (shown in **Fig. R3**) was included as **Suppl. Fig. 6a**.

9. In Figure S3, the molecular weight ladder needs to be labeled. **Done!**

Page 3 lines 71-73. A consistent approach to addressing RCC1 modifications would be appreciated, annotate positions 2-3 within the text with 3 letter residue codes. **Done!**

Page 4 Lines 95-97. This is a confusing sentence! I would suggest to split up the sentence into two sentences: a full sentence discussing the cell contents and the large degrees of flexibility observed, and then address how this relates to the challenge of refining this low-res structure and the respectable refinement statistics achieved. **Done!** We split the sentence into two (page 4, line 99): “**Flexible torsion-angle non-crystallographic symmetry restrains (4) among the eight copies in the asymmetric unit were used throughout refinement. This yielded an accurate atomic model ($R_{work}/R_{free} \sim 27.8/29.6$) (Suppl. Table 1), despite the modest resolution of crystallographic data ($\sim 3.45 \text{ \AA}$)**”.

Also, the range of conformations shown in Figure S1 in main text Figure 5 should be discussed. **Done!** Fig. S1 is discussed in the main text at lines 112-115. To better discuss Fig. 5, we added the following sentence at page 8 (line 233): “**In this ensemble model, the eight β -propellers occupy nearly identical positions ($RMSD 0.26 \text{ \AA}$), while the RCC1 tails are displaced up to 10 \AA with $RMSD \sim 3.1 \text{ \AA}$ between residues 1-27**”.

Page 15 onwards refers to a ‘Complex buffer’ which is subsequently termed ‘Gel filtration’ buffer. Consider consolidating the nomenclature to a single name for each unique buffer used. The two buffers are similar but not identical: **Complex buffer** = 20mM Tris-HCl pH 8.0, 150 mM NaCl, 1 mM EDTA, 5 mM BME, 0.1 mM PMSF; **Gel Filtration (GF) buffer** = 20 mM Tris-HCl pH 8.0, 150 mM NaCl, 5 mM BME, 0.1 mM PMSF.

The reported crystal structures, in light of their lower resolution, would benefit from additional validation using Mol-Probity. I would suggest to include the Mol-Probity score and refinement clash scores in Table S1. **Done!** The Clashscore is 8.8 and 11.6 for importin $\alpha 3$:RCC1 and Kap60:yRCC1, respectively.

REVIEW 3

1. For RCC1, the quality of the original data raises complications, which should be addressed in the manuscript. Nevertheless, the interpretation of the data seems acceptable. **Done!** Guinier plots for all three concentrations of RCC1 (e.g. 2.5, 3.0 and 5.0 mg ml⁻¹) are now shown in **Suppl. Fig. 2b**. RCC1 tends to aggregate at higher concentration, which explains why we carried out *ab initio* modeling using data at the lowest concentration (2.5 mg ml⁻¹).

2. For RCC1-Kap60 complex, the quality of the original data is dubious. For this system, a SEC-SAXS experiment is most likely necessary to avoid interpreting artifacts. As it is, the data from RCC1-Kap60 is not suited for publication. **Done!** We carried out a SEC-SAXS experiment at CHESS in collaboration with Dr. Richard Gillilan (now a co-author of this paper). The new SEC-SAXS data are shown in **Fig. 6c-d** and **Suppl. Fig. 2c**. We thank the reviewer for suggesting this technique, which is very powerful.

3. In general, the authors' representation of the SAXS data is rather unconventional. I would recommend following the guidelines of the IUCr for reporting BioSAXS data, which can be found here: <http://journals.iucr.org/d/issues/2012/06/00/be5200/index.html> **Done!** We added several data collection and analysis parameters to **Suppl. Table 2**.

4. The authors use the term "gyration radius (Gr)" in place of the considerably more common "radius of gyration (RG)". **Done!** We changed all "gyration radius (Gr)" to "radius of gyration (RG)".

5. Referring to the high resolution, crystallographic models as X-ray models is not good nomenclature, as the *ab initio* models derived from the SAXS data are also "X-ray" models. Crystallographic models would be a more suitable term. **Done!** We changed all "X-ray model" to "crystallographic model".

P 5: SAXS analysis of the RCC1-importin complex:

a.) The authors state that several concentrations were measured and that further analysis was performed for all concentrations, but do not describe how robust their findings were and which data was used for the model fitting and figures. It should be noted that performing individual analysis at each concentration, instead of performing extrapolation to zero concentration, is a very unusual approach. We repeated all SAXS experiments at several concentrations, using different buffers. In the case of the importin $\alpha 3$:RCC1 complex, we collected SAXS data at 3.5, 5.0 and 7.5 mg ml⁻¹, and the relative Guinier plots are now shown in **Suppl. Fig. 2a**. We used merged data from these 3 concentrations to calculate an *ab initio* shape reconstruction. Also, we show the normalized Kratky plot for merged data in **Suppl. Fig. 2a**.

b.) In addition, the concentrations reported in the text and those in figure S2 do not match. While the χ^2 of the FOXS fit is indeed good, the figure 2c shows some deviations. The term "excellent" is therefore a bit exaggerated. In particular, the fact that both RG and Dmax of the predicted curves are smaller than observed is a better indicator for the postulated elongation than the comparison of *ab initio* SAXS models and crystallographic models. Given these deviations, they authors should consider using more advanced modeling tools such as sreflex from the ATSAS package to confirm that an elongation would result in a match between predicted and measured SAXS data. **Done!** Both RG and Dmax calculated from SAXS data describe the solvated state of a macromolecule, whereas the maximum dimension of a crystallographic model is measured from 'sidechain-to-sidechain' excluding the solvation layer. Therefore, a difference of <5% between SAXS data and crystallographic model is – indeed – reasonable. Nonetheless, we changed "Excellent" to "**very good**". Finally, we tried modelling the $\alpha 3$:RCC1 complex using sreflex, which – however – severely distorted the importin $\alpha 3$ structure breaking its helical packing between Armadillo repeats 3-4.

c.) The comparison between the predicted and the measured SAXS curves shows some significant deviations, both in $I(q)$ vs. q and the $p(r)$ functions. Judging from the $p(r)$ function this might be due to a small, but not negligible presence of higher oligomers. Despite these issues, based on the data shown, the conclusion that the crystallographic model and the SAXS data agree well seems correct. The reviewer is correct, but fortunately the variations are fairly small, e.g. less than 5Å in max dimension in the $P(r)$ function. In the revised paper, we also provide evidence all samples used for SAXS are both pure and monodisperse as assessed by Analytical Ultracentrifugation Sedimentation Velocity (AUC-SV) and Size Exclusion Chromatography (SEC) (see the new **Suppl. Figure 7** and **Suppl. Table 2**), although we cannot rule out that small aggregates form during SAXS analysis as a result of X-ray irradiation.

SAXS analysis of RCC1:

a.) As previously, it is not clear how the protein concentration affects the results of the analysis. Again, the concentrations reported in the text and those in figure S2 do not match. We corrected the concentration reported in the text (Page 8, line 213): “SAXS analysis in a concentration range between 2.5 and 5.0 mg ml⁻¹”. Guinier plots for all concentrations are now shown in **Suppl. Fig. 2b**.

b.) As figure S2B clearly shows a concentration dependence of RG, the statement that the data is consistent with a monomer cannot be generally true. Both AUC-SV and SEC (see the new **Suppl. Figure 7** and **Suppl. Table 2**) confirmed RCC1 exists as a monodisperse and homogenous monomer in a concentration range between 0.8-2.5 mg ml⁻¹ (see **Suppl. Figure 7** and **Suppl. Table 2**). However, as the reviewer points out, SAXS clearly revealed RCC1 tends to aggregate above ~2.5 mg ml⁻¹, which explains why we used only low concentration data for *ab initio* modelling.

Figure 5, in particular the P(r) function, clearly shows larger objects than the crystallographic model contributing to the SAXS data. In addition with the concentration dependence of the radius of gyration, this raises the question to which extent the racket handle is an artifact due to higher oligomers or aggregation. This caveat of the interpretation should be mentioned in the manuscript. We believe the racket-like shape of the full length RCC1 suggested by SAXS is a genuine description of this molecule. In this respect, hydrodynamic analysis by AUC-SV reveals a frictional ratio $f/f_0 \sim 1.7$ (**Suppl. Figure 7d-e** and **Suppl. Table 2**), also consistent with an elongated macromolecule. We added the following sentence at page 8, line 223: “**The elongated shape suggested by SAXS is also validated by hydrodynamic analysis using Analytical Ultracentrifugation Sedimentation Velocity (AUC-SV) (Suppl. Table 2) that, at comparable concentration, yields a frictional ratio $f/f_0 \sim 1.7$, indicative of an elongated macromolecule.**”

P9: SAXS analysis of RCC1-Kap60 complex:

a.) As for the previous two samples, it is not clear how the protein concentration affects the results of the analysis, and the concentrations and radii of gyrations reported in the text and those in figure S2 do not match.

b.) In particular the statement that the samples are monodisperse below 2.5 mg/mL seems dubious, as the SAXS signal apparently is not concentration independent in this range.

c.) The statement that “An *ab initio* shape reconstruction from scattering data at 1, 2 and 2.5 mg ml⁻¹ reveals an elongated shape (Fig. 6d), similar to a boomerang, in reasonable agreement with the X-ray structure ($\chi^2 \sim 3.0$)” seems to imply that the χ^2 of the *ab initio* model is 3.0, which seems unlikely. It should be clearly stated that the SAXS data was compared to predicted curves from a crystallographic model.

d.) As a matter of fact, the predicted and measured SAXS do not match at all. As the SAXS data seems to have some aggregates from higher oligomers/aggregates, this is not surprising.

RCC1-Kap60 probably needs to be measured with SEC-SAXS to address the issue of sample monodispersity.

We thank the reviewer for suggesting SEC-SAXS. We were lucky to obtain beamtime at CHESS, where we performed a SEC-SAXS experiment in collaboration with Dr. Richard Gillilan. The new SEC-SAXS data, illustrated in **Fig. 6c-d**, **Suppl. Fig. 2c** and **Suppl. Table 2**, are significantly improved as compared to the previous conventional SAXS (see comparison in **Fig. R4**). Although there is a small percentage of a larger aggregate in solution possibly responsible for the differences between crystallographic model and solution data ($\chi^2 = 2.15$), the new SEC-SAXS data are considerably more similar to the crystallographic model. Furthermore, AUC-SV confirmed the Kap60:yRCC1 complex sediments as a single 98.2 kDa species (though with a broader distribution than the importin $\alpha 3$:RCC1 complex, see **Suppl. Figure 7h vs 7b**) that corresponds to a 1:1 heterodimer of Kap60 and yRCC1. The M.W. extrapolated from SAXS (99.0 kDa) and AUC-SV (98.0 kDa) are indeed very similar to the theoretical M.W. of 103.0 kDa

(Suppl. Table 2). In contrast, SEC, which is shape-biased, suggests the Kap60:yRCC1 complex migrates like a ~175 kDa species (Suppl. Figure 7I) that may reflect an elongated 1:1 complex possibly in equilibrium with a dimeric species. To describe the new SEC-SAXS data, we completely re-wrote the Results at page 8 (line 252) and Methods at page 17 (line 512), and revised Figs. 6c-d.

P16: Methods of SAXS analysis and ab initio shape reconstruction:

- a.) line 486: The composition of the gel filtration buffer is never explicitly stated. Is it the same as the complex buffer described on page 15, line 423? The composition of the Gel Filtration buffer (GF buffer) is stated at page 14, line 451. It consists of: 20mM Tris-HCl pH 8.0, 150 mM NaCl, 5 mM BME, 0.1 mM PMSF.
- b.) Line 471 - 474: Which software was used for data reduction and Guinier analysis? It also seems very probable that data reduction was performed prior to the Guinier analysis. The Rigaku Automatic Data Analysis Pipeline software and BioSAXS-RAW software (5) were used for data reduction while Guinier analysis was performed using RAW (5), Scatter (6) and TSAS software (7). See revised Suppl. Table 2.
- c.) Line 479: I would assume that FoXS was used to calculate all theoretical scattering curves in the manuscript, not only the one of importin α 3-RCC1 as stated. Yes, we added the following sentence at page 17 (line 524): "**Theoretical solution scattering curves of all crystallographic structures determined in this work were calculated using the FoXS web server**".
- d.) Line 480: Instead of suppl. table 1, suppl. table 2 should be referenced here. However, table 2 contains no data collection statistics. **Done!** In addition, we added SAXS data collection statistics to Suppl. Table 2 and also added AUC-SV statistics. This comparative table is now named: "**Summary of biophysical parameters measured by SAXS and AUC**".
- e.) This section (nor any other part of the manuscript) contains now reference whatsoever on how the monodispersity of the SAXS data was assessed. As the interpretation of the data depends strongly on the monodispersity, this is a major flaw. **Done!** As previously mentioned, we tested the monodispersity of all samples used for SAXS using AUC-SV and SEC. We added the following sentence at the beginning of the SAXS Methods (page 16, line 498): "**The monodispersity of all samples used for SAXS analysis was validated using Analytical Ultracentrifugation Sedimentation Velocity (AUC-SV) and Size Exclusion Chromatography (SEC) (Suppl. Figure 7)**".

P. 22: Caption for Figure 2 & P23: Captions for Figure 5 & 6:

All SAXS data, including those in panel c) should be clearly identified (concentration!). **Done!** We rewrote the legends of Fig. 2c,d; Fig. 5a,b and Fig. 6c,d and included concentrations.

- a.) Figure 2c, 5a and 6c: The y-axis label on the $I(q)$ vs. q plot should be $\log(I(q))$ not $I(q)$. **Done!** In figure 2c, 5a and 6c we show buffer-subtracted one-dimensional scattering data that we plot as **Log $I(q)$ vs $q(\text{\AA}^{-1})$** . Relative Guinier plots expressed as **Log $I(q)$ vs $q^2(\text{\AA}^{-2})$** are presented in Suppl. Figure 2.

Figure S2:

- a.) The presented concentrations do not match with those presented in the relevant sections of the main body of the manuscript. **Done!** Fixed
- b.) This figure is the only place in the manuscript where the authors use the abbreviation RG for the radius of gyration. **Done!** We now uniformly refer to the radius of gyration as RG
- c.) The radius of gyration in the RG vs concentration plots requires units. **Done!** We revised this to: RG (\AA)
- d.) The Kratky plots should be presented normalized for RG. Also, it should be stated to which concentration they belong. **Done!** Revised normalized Kratky plots are provided in Suppl. Fig. 2. Relevant concentrations are mentioned in the figure legends.
- e.) Guinier plots for all presented SAXS data should be added to this figure. **Done!**

Suppl. table 2: Contrary to its caption the table contains no information on the SAXS data collection. **Done!** We added data collection and model statistics to Suppl. Table 2. This table also contains additional biophysical parameters from AUC-SV such as the Sedimentation Coefficient s , Radius of Hydration (RH) and frictional ratio (f/f_0).

References

1. Chen, V. B., Arendall, W. B., 3rd, Headd, J. J., Keedy, D. A., Immormino, R. M., Kapral, G. J., Murray, L. W., Richardson, J. S., and Richardson, D. C. (2010) MolProbity: all-atom structure validation for macromolecular crystallography. *Acta Crystallogr D Biol Crystallogr* **66**, 12-21
2. Renault, L., Nassar, N., Vetter, I., Becker, J., Klebe, C., Roth, M., and Wittinghofer, A. (1998) The 1.7 Å crystal structure of the regulator of chromosome condensation (RCC1) reveals a seven-bladed propeller. *Nature* **392**, 97-101
3. Makde, R. D., England, J. R., Yennawar, H. P., and Tan, S. (2010) Structure of RCC1 chromatin factor bound to the nucleosome core particle. *Nature* **467**, 562-566
4. Headd, J. J., Echols, N., Afonine, P. V., Moriarty, N. W., Gildea, R. J., and Adams, P. D. (2014) Flexible torsion-angle noncrystallographic symmetry restraints for improved macromolecular structure refinement. *Acta Crystallogr D Biol Crystallogr* **70**, 1346-1356
5. Nielsen, S. S., Toft, K. N., Snakenborg, D., Jeppesen, M. G., Jacobsen, J. K., Vestergaard, B., Kutter, J. P., and Arleth, L. (2009) BioXTAS RAW, a software program for high-throughput automated small-angle X-ray scattering data reduction and preliminary analysis. *Journal of Applied Crystallography* **42**, 959-964
6. Förster, S., Apostol, L., and Bras, W. (2010) Scatter: software for the analysis of nano- and mesoscale small-angle scattering. *J. Appl. Cryst.* **43**, 639-646
7. Petoukhov, M. V., Franke, D., Shkumatov, A. V., Tria, G., Kikhney, A. G., Gajda, M., Gorba, C., Mertens, H. D., Konarev, P. V., and Svergun, D. I. (2012) New developments in the ATSAS program package for small-angle scattering data analysis. *J Appl Crystallogr* **45**, 342-350

REVIEWERS' COMMENTS:

Reviewer #1 (Remarks to the Author):

The authors have done an excellent job addressing the reviewers' concerns. I have no further comments and recommend publication of this highly interesting study.

Reviewer #2 (Remarks to the Author):

The revised manuscript presented by Sankhala et al has addressed all of my concerns. I recommend that the manuscript will be published without further delay. I congratulate the authors to a very exciting story.

Reviewer #3 (Remarks to the Author):

I would like to thank the authors for their efforts which improved the quality of the manuscript. The SAXS data is now presented much more convincingly. I especially appreciate the direct comparison to the AUC data.

There are just a few very minor issues to be fixed:

General remarks:

It is admittedly a bit nitpicky, but in the log Intensity vs q plots, the y-axis title should be $I(q)$ (a.u.), as the tick labels correspond to the intensity and not its logarithm.

For both the SAXS data and the $p(r)$ functions (main text and supplement figures), error bars should be shown.

Errors for derived values, such as R_G should be provided, esp. in the supplement figures.

P9, line 265:

"An ab initio shape reconstruction from SEC-SAXS data reveals an elongated shape (Fig. 6d), in reasonable agreement with the crystallographic structure ($\chi^2 \sim 2.15$)."

This sentence seems a bit misleading. What is the χ^2 value referring to? The FoXS fit?

P 17, line 513:

I am a bit surprised by the Finger Lakes CCD detector. According to this page (http://www.macchess.cornell.edu/MacCHESS/beamline_character.html) G1 has a dual Pilatus 100K-S SAXS/WAXS setup.

Response to Reviewer #3

I would like to thank the authors for their efforts which improved the quality of the manuscript. The SAXS data is now presented much more convincingly. I especially appreciate the direct comparison to the AUC data.

Thank-you!

There are just a few very minor issues to be fixed:

1. It is admittedly a bit nitpicky, but in the log Intensity vs q plots, the y-axis title should be I(q) (a.u.), as the tick labels correspond to the intensity and not its logarithm.

Done! The Y-axis label has been fixed as per the reviewer's suggestion.

2. For both the SAXS data and the p(r) functions (main text and supplement figures), error bars should be shown.

Done! We updated the Iq and Pr function plots in Figs 2, 5 and 6 by incorporating error bars. Because the error bars are very small, we used a solid line instead of circles to represent SAXS data.

3. Errors for derived values, such as RG should be provided, esp. in the supplement figures.

Done! e.g. RGs = 34.6 ± 0.9 ; 27.8 ± 0.7 ; 45.9 ± 0.5 . See Supplementary Table 1 and main text.

4. P9, line 265:

“An ab initio shape reconstruction from SEC-SAXS data reveals an elongated shape (Fig. 6d), in reasonable agreement with the crystallographic structure ($\chi^2 \sim 2.15$).”

This sentence seems a bit misleading. What is the chi2 value referring to? The FoXS fit?

Yes. The Chi2 value (obtained from FoXS) refers to the fitting of the crystallographic model against the observed SAXS data.

5. P 17, line 513:

I am a bit surprised by the Finger Lakes CCD detector. According to this page

(http://www.macchess.cornell.edu/MacCHESS/beamline_character.html) G1 has a dual Pilatus 100K-S SAXS/WAXS setup.

The Reviewer is correct. Data were collected on a dual Pilatus 100K-S detector.